# LORD: LOWER-DIMENSIONAL EMBEDDING OF LOG-SIGNATURE IN NEURAL ROUGH DIFFERENTIAL EQUATIONS

**Jaehoon Lee, Jinsung Jeon, Sheoyon Jhin, Jihyeon Hyeong, Jayoung Kim,
Minju Jo, Seungji Kook, and Noseong Park**
Yonsei University
Seoul, South Korea
{jaehoonlee,jjsjjs0902,sheoyonj,jiji.hyeong,jayoung.kim,
alflsowl12,2021321393,noseong}@yonsei.ac.kr

## ABSTRACT

The problem of processing very long time-series data (e.g., a length of more than 10,000) is a long-standing research problem in machine learning. Recently, one breakthrough, called neural rough differential equations (NRDEs), has been proposed and has shown that it is able to process such data. Their main concept is to use the log-signature transform, which is known to be more efficient than the Fourier transform for irregular long time-series, to convert a very long time-series sample into a relatively shorter series of feature vectors. However, the log-signature transform causes non-trivial spatial overheads. To this end, we present the method of **LO**we**R**-**D**imensional embedding of log-signature (LORD), where we define an NRDE-based autoencoder to implant the higher-depth log-signature knowledge into the lower-depth log-signature. We show that the encoder successfully combines the higher-depth and the lower-depth log-signature knowledge, which greatly stabilizes the training process and increases the model accuracy. In our experiments with benchmark datasets, the improvement ratio by our method is up to 75% in terms of various classification and forecasting evaluation metrics.

## 1 INTRODUCTION

Time-series data occurs frequently in real-world applications, e.g., stock price forecasting (Ariyo et al., 2014; Siami-Namini et al., 2018; Jhin et al., 2021), traffic forecasting (Reinsel, 2003; Fu, 2011; Bai et al., 2020; Fang et al., 2021), weather forecasting (Shi et al., 2015; Seo et al., 2018; Brouwer et al., 2019; Ren et al., 2021), and so on. However, it is known that very long time-series data (e.g., a time-series length of more than 10,000) is not straightforward to process with deep learning despite various techniques ranging from recurrent neural networks (RNNs) to neural ordinary/controlled differential equations (NODEs and NCDEs). RNNs are known to be unstable when training with such very long sequences and the maximum length that can be processed by NODEs and NCDEs is more or less the same as that by RNNs (Morrill et al., 2021; Aicher et al., 2020; Trinh et al., 2018; Stoller et al., 2019; Bai et al., 2018). However, one breakthrough has been recently proposed, namely neural rough differential equations (NRDEs).

NRDEs are based on the rough path theory which was established to make sense of the controlled differential equation:

$$d\mathbf{z}(t) = f(\mathbf{z}(t))dX(t), \tag{1}$$

where $X$ is a continuous control path, and $\mathbf{z}(t)$ is a hidden vector at time $t$. A prevalent example of $X$ is a (semimartingale) Wiener process, in which case the equation reduces to a stochastic differential equation. In this sense, the rough path theory extends stochastic differential equations beyond the semimartingale environments (Lyons et al., 2004).

One key concept in the rough path theory is the *log-signature* of a path. It had been proved that the log-signature of a path with bounded variations is unique under mild conditions (Lyons & Xu,

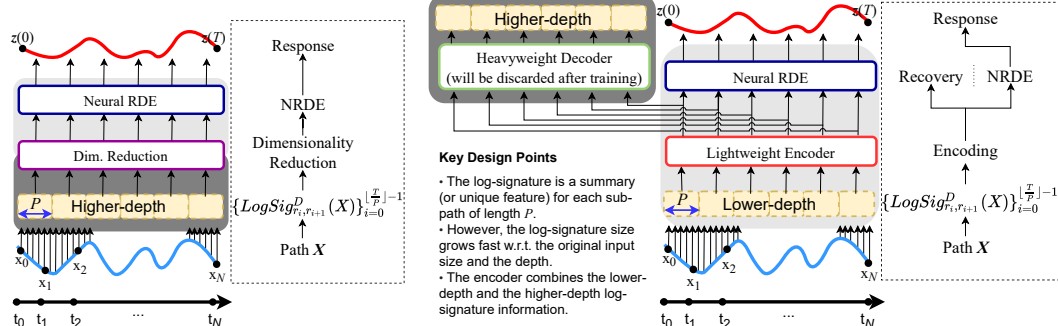

(a) Directly reducing the dimensionality      (b) Combining both log-signature information (our method)

Figure 1: Two possible approaches to reduce the dimensionality of the higher-depth log-signature. The dark gray means a processing in a higher-dimensional space and the light gray means that in a lower-dimensional space. Our embedding method in (b) is better than the baseline in (a) in that i) the higher-dimensional processing is deferred to the decoder which will be discarded after pre-training the autoencoder, and ii) our method does not involve any higher-dimensional processing during the main training and inference processes. We pre-train the autoencoder and in the main training step, we discard the decoder, fix the encoder, and train only the main NRDE for a downstream task (cf. Table 1). In this way, we can exclude the higher-dimensional processing as early as possible.

2018; Geng, 2017) and most time-series data that happens in the field of deep learning has bounded variations. Therefore, one can interpret that the log-signature is a unique feature of the path.

Given a $N$-length time-series sample $\{\mathbf{x}_i\}_{i=0}^N$ annotated with its observation time-points $\{t_i\}_{i=0}^N$, where $t_0 = 0$, $t_N = T$, and $t_i < t_{i+1}$, NRDEs construct a continuous path $X(t)$, where $t \in [0, T]$, with an interpolation algorithm, where $X(t_i) = (\mathbf{x}_i, t_i)$ for $t_i \in \{t_i\}_{i=0}^N$. In other words, the path has the same value as the observation $(\mathbf{x}_i, t_i)$, when $t_i$ is one of the observation time-points and otherwise, interpolated values. As shown in Fig. 1, a log-signature (each dotted yellow box in the figure) is calculated every $P$-length sub-path, and another time-series of log-signatures, denoted $\{LogSig_{r_i, r_{i+1}}^D(X)\}_{i=0}^{\lfloor \frac{T}{P} \rfloor - 1}$, is created. The log-signature calculation has one important hyperparameter $D$ called *depth* — the higher the depth is, the more accurately represented each sub-path is (cf. Eq. 4). For instance, the best accuracy score is 0.81 for $D = 3$ vs. 0.78 for $D = 2$ in EigenWorms. The sub-path length $P$ and the depth $D$ decides the number and the dimensionality of log-signatures, respectively.

However, one downside is $\dim(LogSig_{r_i, r_{i+1}}^D(X)) > \dim(X)$, where $\dim(X)$ means the dimensionality (or the number of channels) of $X$. As a matter of fact, $\dim(LogSig_{r_i, r_{i+1}}^D(X))$ grows rapidly w.r.t. $\dim(X)$ (Morrill et al., 2021). Since the dimensionality of the input data is, given a dataset, fixed, we need to decrease $D$ to reduce overheads. In general, NRDEs require more parameters to process higher-dimensional log-signatures (as shown in Table 4 where the original NRDE design always requires more parameters for $D = 3$ in comparison with $D = 2$.)

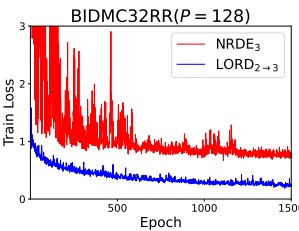

Figure 2: The loss curve of LORD is stable. Other figures are in Appendix F.

To this end, we propose to embed the higher-depth signatures onto a lower-dimensional space to i) decrease the complexity of the main NRDE (the solid blue box in Fig. 1), and ii) increase the easiness of training (cf. Fig. 2) and as a result, its model accuracy as well. Fig. 1 shows two possible approaches where we directly reduce the dimensionality of the higher-depth log-signature in Fig. 1 (a) or we adopt the autoencoder architecture to combine both the higher-depth and the lower-depth signatures in Fig. 1 (b) — the first approach in Fig. 1 (a) is a baseline model, and our proposed model is the second approach in Fig. 1 (b). Autoencoders are frequently used for (unsupervised) dimensionality reduction although there also exist other approaches. Since the higher-depth log-signature should be reconstructed from the embedded vector, one can consider that

the higher-depth log-signature is indirectly embedded into the vector. Moreover, the decoder is discarded after its pre-training so our method does not involve any high-cost computation with the higher-depth log-signature in the main training and inference steps as clarified in Table 1.

Table 1: Two phase training in our method

|  | Pre-training | Main training |
| --- | --- | --- |
| Decoder | Training | Discarded |
| Encoder | Training | Fixed |
| Main NRDE | Not Applicable | Training |

Our proposed method, **LO**we**R**-**D**imensional embedding of log-signature (LORD), adopts an NRDE-based autoencoder to combine the higher-depth and the lower-depth log-signature knowledge, and utilizes the embedded knowledge by the encoder for a downstream machine learning task as shown in Fig. 1 (b). The encoder embeds the lower-depth log-signature from the continuous path $X$, and the decoder reconstructs the higher-depth log-signature. This specific design is where our key idea lies in. The encoded lower-depth log-signature will contain both the lower-depth and the higher-depth log-signature knowledge because i) it is an encoding of the lower-depth log-signature and ii) the decoder should recover the higher-depth log-signature from it. After pre-training the autoencoder, we discard the decoder to exclude the higher-dimensional processing as early as possible from our framework. After that, we fix the encoder and train only the main NRDE.

We conduct experiments with six very long time-series benchmark datasets and five baselines including NODE, NCDE, and NRDE-based models. RNN-based models cannot process the very long time-series datasets and we exclude them. Our proposed method significantly outperforms all existing baselines. Our contributions can be summarized as follows:

1. We design an NRDE-based *continuous* autoencoder to combine the higher-depth and the lower-depth log-signature information. Since the decoder is discarded after being trained, we adopt the asymmetric architecture that the encoder is lightweight and the decoder is heavyweight in terms of the number of parameters.

2. Our proposed method outperforms all baselines by large margins (up to 75% for various evaluation metrics, e.g., $R^2$) in standard benchmark datasets with much smaller model sizes in comparison with the original NRDE design.

Our code is available in `https://github.com/leejaehoon2016/LORD`.

## 2 RELATED WORK AND PRELIMINARIES

**NODEs** NODEs (Chen et al., 2018) use the following equation:

$$\mathbf{z}(T) = \mathbf{z}(0) + \int_0^T f(\mathbf{z}(t), t; \theta_f)dt, \tag{2}$$

which means $\mathbf{z}(T)$ is solely defined by the initial value $\mathbf{z}(0)$ — the entire evolving path from $\mathbf{z}(0)$ to $\mathbf{z}(T)$ is defined by the initial value in ODEs. NODEs are not considered as a continuous analogue to RNNs but to residual networks (Chen et al., 2018).

**NCDEs** Let $\{\mathbf{x}_i\}_{i=0}^N$ be an $N$-length time-series sample and $\{t_i\}_{i=0}^N$ be its observation time-points. NCDEs are different from NODEs in that they use the following equation:

$$\mathbf{z}(T) = \mathbf{z}(0) + \int_0^T f(\mathbf{z}(t); \theta_f)dX(t) = \mathbf{z}(0) + \int_0^T f(\mathbf{z}(t); \theta_f)\frac{dX(t)}{dt}dt, \tag{3}$$

where $X(t)$ is a continuous interpolated path from $\{(\mathbf{x}_i, t_i)\}_{i=0}^N$, where $t_N = T$. In this regards, NCDEs can also be considered as a continuous analogue to RNNs. The adjoint sensitivity method can be used to decrease the space complexity of gradient calculation instead of backpropagation (Kidger et al., 2020).

However, NCDEs has an inherent disadvantage that they are weak at processing very long time-series samples, because NCDEs directly process the very long time-series samples.

**Signature Transform**  Let $[r_i, r_{i+1}]$ be a time duration, where $r_i < r_{i+1}$. The signature of the time-series sample is defined as follows:

$$S_{r_i,r_{i+1}}^{i_1,\ldots,i_k}(X) = \int\cdots\int_{r_i<t_1<\ldots<t_k<r_{i+1}} \prod_{j=1}^{k} \frac{dX^{i_j}}{dt}(t_j)dt_j,$$

$$Sig_{r_i,r_{i+1}}^{D}(X) = \Big(\{S_{r_i,r_{i+1}}^{i}(X)\}_{i=1}^{\dim(X)}, \{S_{r_i,r_{i+1}}^{i,j}(X)\}_{i,j=1}^{\dim(X)}, \ldots, \{S_{r_i,r_{i+1}}^{i_1,\ldots,i_D}(X)\}_{i_1,\ldots,i_D=1}^{\dim(X)}\Big),$$

$$(4)$$

However, the signature has redundancy, e.g., $S_{a,b}^{1,2}(X) + S_{a,b}^{2,1}(X) = S_{a,b}^{1}(X)S_{a,b}^{2}(X)$ where we can know a value if knowing three others. The log-signature $LogSig_{r_i,r_{i+1}}^{D}(X)$ is then created after dropping the redundancy from the signature. In this work, we use this log-signature definition in default to design our method. By creating a log-signature for every $P$-length sub-path, the entire sequence length can be reduced — moreover, the log-signature is a unique feature of the sub-path (Lyons & Xu, 2018; Geng, 2017). For complexity reasons, we typically use the log-signature of depth $D \le 3$ (Morrill et al., 2021).

**NRDEs**  Owing to the log-signature transform of time-series, NRDEs (Morrill et al., 2021) are defined as follows:

$$g(X,t) = \frac{LogSig_{r_i,r_{i+1}}^{D}(X)}{r_{i+1} - r_i} \text{ for } t \in [r_i, r_{i+1}),$$

$$\mathbf{z}(T) = \mathbf{z}(0) + \int_{0}^{T} f(\mathbf{z}(t); \theta_f)g(X,t)dt,$$

$$(5)$$

where $LogSig_{r_i,r_{i+1}}^{D}(X)$ means the log-signature created from the path $X$ within the interval $[r_i, r_{i+1})$. $\frac{LogSig_{r_i,r_{i+1}}^{D}(X)}{r_{i+1}-r_i}$ is a piecewise approximation of the time-derivative of the log-signature in the short interval $[r_i, r_{i+1})$. $D$ means the depth of the log-signature. Once we define the sub-path length $P$, the intervals $\{[r_i, r_{i+1})\}_{i=0}^{\lfloor\frac{T}{P}\rfloor-1}$ are decided, i.e., $P = r_{i+1} - r_i$ for all $i$. Then, the time-series of $LogSig_{r_i,r_{i+1}}^{D}(X)$ constitutes $\{LogSig_{r_i,r_{i+1}}^{D}(X)\}_{i=0}^{\lfloor\frac{T}{P}\rfloor-1}$ in our notation.

NRDEs use Eq. 5 to derive $\mathbf{z}(T)$ from $\mathbf{z}(0)$, which can be considered as a continuous analogue to RNNs since it continuously reads the time-derivative of the log-signature. Therefore, $\mathbf{z}(T)$ is defined by the initial value $\mathbf{z}(0)$ and the sequence of the time-derivative of the log-signature. We can also use the adjoint sensitivity method to train NRDEs.

**Long Sequence Time-series Input (LSTI)**  The problem of processing very long time-series data is a long standing research problem. There are three ways to solve the LSTI problem. First, it reduces the sequence through truncating/summarizing/sampling from a very long input sequence. However, this method may lose information affecting the prediction accuracy. The second is to give the gradient transformation. As the sequence becomes longer, the gradient vanishing problem occurs. Therefore, in (Aicher et al., 2020), the model is trained using only the gradient of the last step. It also solves the problem using auxiliary losses (Trinh et al., 2018). Finally, CNNs (Stoller et al., 2019; Bai et al., 2018) were used to solve the LSTI problem. The convolutional filter captures long term dependencies, but the receptive fields increase exponentially, breaking the sequence.

## 3  PROPOSED METHOD

We describe our proposed method, LORD-NRDE. We first clarify the motivation of our work and then describe our proposed model design.

**Motivation**  It is known that NRDEs are a generalization of NCDEs (Morrill et al., 2021, Section 3.2). However, one problem in utilizing NRDEs in real-world environments is that $\dim(LogSig_{r_i,r_{i+1}}^{D}(X))$ is a rapidly growing function of $\dim(X)$ (Morrill et al., 2021, Section A).

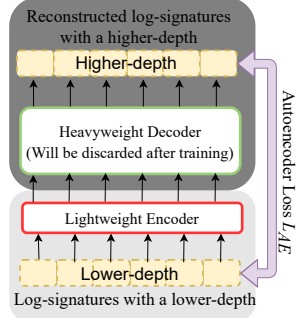

Figure 3: The proposed NRDE-based autoencoder

**Algorithm 1:** How to train LORD-NRDE

**Input:** Training data $D_{train}$, Validating data $D_{val}$, Maximum iteration numbers $max\_iter_{AE}$ and $max\_iter_{TASK}$

1 Initialize the parameters of the encoder (i.e., $\theta_f$, $\theta_{\phi_{\mathbf{e}}}$), the decoder (i.e., $\theta_o$, $\theta_{\phi_{\mathbf{s}}}$), and the main NRDE (i.e., $\theta_g$, $\theta_{\phi_{\mathbf{z}}}$, $\theta_{\phi_{\mathbf{output}}}$).;

    /* Pre-training of the autoencoder        */

2 **for** $max\_iter_{AE}$ *iterations* **do**

3     | Train the encoder and the decoder using $L_{AE}$ ;

    /* Main-training of LORD-NRDE        */

4 **for** $max\_iter_{TASK}$ *iterations* **do**

5     | Train the main NRDE using $L_{TASK}$;

6     | Validate the best main NRDE parameters with $D_{val}$;

7 **return** the encoder and the main NRDE parameters;

This rapid blow-up of dimensionality causes two problems : i) it hinders us from applying NRDEs to high-dimensional time-series data, and ii) it makes the training process complicated since the hidden representation of the large input should be made for a downstream task. Therefore, we propose to embed the log-signature onto a lower dimensional space so that the training process becomes more tractable and enhance the applicability of NRDEs to high-dimensional data. After the embedding, in other words, the dimensionality of embedded vector is typically smaller than that of the log-signature with depth $D_2$ in our setting but it has knowledge of the log-signature with depth $D_2 > D_1$.

**Overall Architecture** We describe our proposed method, LORD-NRDE, which consists of three NRDEs: i) an encoder NRDE, ii) a decoder NRDE and iii) a main NRDE to derive $\mathbf{z}(T)$ from $\mathbf{z}(0)$. The role of each NRDE is as follows:

1. The encoder NRDE *continuously* embeds the log-signature with depth $D_1$ onto another space. We use $\mathbf{e}(t)$ to denote the vector embedded from the log-signature at time $t$.
2. The decoder NRDE reconstructs the log-signature with depth $D_2 > D_1$ from $\mathbf{e}(t)$.
3. The main NRDE reads $\mathbf{e}(t)$ to evolve $\mathbf{z}(t)$ and derive $\mathbf{z}(T)$. There is an output layer which reads $\mathbf{z}(T)$ and makes inference.

We note that $\dim(LogSig^{D_2}_{r_i,r_{i+1}}(X)) > \dim(\mathbf{e}(t))$, where $t \in [0, T]$. We require that the encoder produces embedding vectors that contain both the log-signature knowledge with $D_1$ and $D_2$ depth.

**LORD-NRDE** To this end, we propose the following method:

$$\mathbf{z}(T) = \mathbf{z}(0) + \int_0^T g(\mathbf{z}(t); \theta_g) \frac{d\mathbf{e}(t)}{dt} dt, \tag{6}$$

$$\mathbf{e}(T) = \mathbf{e}(0) + \int_0^T f(\mathbf{e}(t); \theta_f) \frac{LogSig^{D_1}_{r_i,r_{i+1}}(X)}{r_{i+1} - r_i} dt, \text{ for } t \in [r_i, r_{i+1}), \tag{7}$$

$$\mathbf{s}(T) = \mathbf{s}(0) + \int_0^T o(\mathbf{s}(t); \theta_o) \frac{d\mathbf{e}(t)}{dt} dt, \tag{8}$$

where $\dim(\mathbf{e}(t)) < \dim(\mathbf{s}(t)) = \dim(LogSig^{D_2}_{r_i,r_{i+1}}(X))$, where $t \in [r_i, r_{i+1})$, with $D_2 > D_1$.

The term $d\mathbf{e}(t)$ can be considered as an embedding of the log-signature with depth $D_1$ at time $t$, and $\mathbf{e}(t)$ as an embedding of $X(t)$ constructed by the log-signature with depth $D_1$. Similarly, $d\mathbf{s}(t)$ is a reconstructed log-signature with depth $D_2$ from the embedding, and $\mathbf{s}(t)$ as $X(t)$ reconstructed by the log-signature with depth $D_2$. Since $\mathbf{e}(0)$ and $\mathbf{s}(0)$ mean the initial values, we set $\phi_{\mathbf{e}}(X(0); \theta_{\phi_{\mathbf{e}}})$, $\phi_{\mathbf{s}}(\mathbf{e}(0); \theta_{\phi_{\mathbf{s}}})$, where $\phi$ means a transformation function.

Therefore, $\mathbf{e}(t)$ and $\mathbf{s}(t)$ constitute an autoencoder based on NDREs as shown in Fig. 3. In our design, we use the asymmetrical setting for them, where the encoder RDE is lightweight and the decoder RDE is heavyweight in terms of the number of parameters, considering the applicability of

our design to real-world environments. Since the decoder NRDE is discarded after its pre-training, the heavyweight setting causes no problems afterwards — our method requires more resources for the pre-training process though.

There are three functions $g$, $f$, and $o$ in Eqs. 6 to 8 and their definitions are as follows:

$$g(\mathbf{z}(t); \theta_g) = \text{FC}_{out_g, N_g+1}(\psi(\text{FC}_{h_g, N_g}(...\eta(\text{FC}_{h_g, 1}(\mathbf{z}(t)))...))), \quad (9)$$

$$f(\mathbf{e}(t); \theta_f) = \text{FC}_{out_f, N_f+1}(\psi(\text{FC}_{h_f, N_f}(...\eta(\text{FC}_{h_f, 1}(\mathbf{e}(t)))...))), \quad (10)$$

$$o(\mathbf{s}(t); \theta_o) = \text{FC}_{out_o, N_o+1}(\psi(\text{FC}_{h_o, N_o}(...\eta(\text{FC}_{h_o, 1}(\mathbf{s}(t)))...))), \quad (11)$$

where $\text{FC}_h$ is a fully-connected (FC) layer whose output size is $h$. $N_g, N_f, N_o$ means the number of FC layers. $\psi$ and $\eta$ are the hyperbolic tangent and rectified linear unit activations, respectively.

**Training Method**    To train the NRDE-based autoencoder, we use the following loss:

$$L_{recon} = \frac{\sum_{i=0}^{M} \|(\mathbf{s}(r_{i+1}) - \mathbf{s}(r_i)) - LogSig_{r_i, r_{i+1}}^{D_2}(X)\|_2^2}{M}, \quad (12)$$

$$L_{AE} = L_{recon} + c_{AE}(\|\theta_f\|_2^2 + \|\theta_o\|_2^2 + \|\theta_{\phi_\mathbf{e}}\|_2^2 + \|\theta_{\phi_\mathbf{s}}\|_2^2) + c_{\mathbf{e}}\|\mathbf{e}\|_2^2, \quad (13)$$

where $M = \lfloor \frac{T}{P} \rfloor - 1$, and $c_{AE}$ is a coefficient of the $L_2$ regularization terms. $c_{\mathbf{e}}$ also regularizes the scale of the learned embedding. In addition to $L_{AE}$, we also have a task loss $L_{TASK}$ which is the standard cross-entropy (CE) loss or the mean squared loss (MSE) loss. From $\mathbf{z}(T)$, we have an output layer, which is typically a fully-connected layer with an appropriate final activation function such as softmax, to produce a prediction $\hat{y}$. $\theta_{\phi_\mathbf{output}}$ denotes the parameters of the output layer. Using the ground-truth information $y$, we define the task loss $L_{TASK}$. Therefore, the final loss can be written as follows:

$$L_{TASK} = CE(y, \hat{y}) + c_{TASK}(\|\theta_g\|_2^2 + \|\theta_{\phi_\mathbf{z}}\|_2^2 + \|\theta_{\phi_\mathbf{output}}\|_2^2), \quad (14)$$

where $c_{TASK}$ is a coefficient of the $L_2$ regularization terms, and $CE(y, \hat{y})$ means a cross-entropy loss — we assume classification but it can be accordingly changed for other tasks.

To implement, we define the following augmented ODE:

$$\frac{d}{dt} \begin{bmatrix} \mathbf{z}(t) \\ \mathbf{e}(t) \\ \mathbf{s}(t) \end{bmatrix} = \begin{bmatrix} g(\mathbf{z}(t); \theta_g) f(\mathbf{e}(t); \theta_f) \frac{LogSig_{r_i, r_{i+1}}^{D_1}(X)}{r_{i+1} - r_i} \\ f(\mathbf{e}(t); \theta_f) \frac{LogSig_{r_i, r_{i+1}}^{D_1}(X)}{r_{i+1} - r_i} \\ o(\mathbf{s}(t); \theta_o) f(\mathbf{e}(t); \theta_f) \frac{LogSig_{r_i, r_{i+1}}^{D_1}(X)}{r_{i+1} - r_i} \end{bmatrix}, \text{ for } t \in [r_i, r_{i+1}), \quad (15)$$

where we replace $\frac{d\mathbf{e}(t)}{dt}$ with $f(\mathbf{e}(t); \theta_f) \frac{LogSig_{r_i, r_{i+1}}^{D_1}(X)}{r_{i+1} - r_i}$ (as defined in Eq. 7), and the initial values are defined as follows:

$$\begin{bmatrix} \mathbf{z}(0) \\ \mathbf{e}(0) \\ \mathbf{s}(0) \end{bmatrix} = \begin{bmatrix} \phi_\mathbf{z}(X(0)); \theta_{\phi_\mathbf{z}}) \\ \phi_\mathbf{e}(X(0)); \theta_{\phi_\mathbf{e}}) \\ \phi_\mathbf{s}(\mathbf{e}(0); \theta_{\phi_\mathbf{s}}) \end{bmatrix}, \quad (16)$$

where $\phi_\mathbf{z}$ and $\phi_\mathbf{e}$ are transformation functions to produce the initial values from the initial observation $X(0)$, and $\phi_\mathbf{s}$ is a transformation function to produce the initial reconstructed value from the initial embedded vector $\mathbf{e}(0)$. We use a fully-connected layer for each of them.

Alg. 1 shows the training algorithm. In Line 3, we pre-train the autoencoder for $max\_iter_{AE}$ iterations. After discarding the decoder and fixing the encoder, we then train the main NRDE in Line 5 for $max\_iter_{TASK}$ iterations. Using $D_{val}$, we validate and choose the best parameters.

## 4  EXPERIMENTAL EVALUATIONS

We describe our experimental environments and results. The mean and variance of 5 different seeds are reported for model evaluation. We refer to Appendix A for reproducibility.

## 4.1 Experimental Environments

**Datasets** We use six real-word dataset which all contain very long time-series samples. There are 3 classification datasets in the University of East Anglia (UEA) repository (Tan & Webb): `EigenWorms`, `CounterMovementJump`, and `SelfRegulationSCP2`, and 3 forecasting datasets in Beth Israel Deaconess Medical Centre (BIDMC) which come from the TSR archive (Tan & Webb): `BIDMCHR`, `BIDMCRR`, `BIDMCSpO2`. We refer to Appendix B for detail of datasets.

**Baselines** There are three types of baselines in our Experiments, ranging from NODE to NCDE and NRDE-based baseline models. `ODE-RNN` is one of the state-of-the-art NODE models in processing time-series. Following the suggestion in (Morrill et al., 2021), we merge $P$ observations into one merged observation and feed it into `ODE-RNN`, in which case `ODE-RNN` is able to process much longer time-series. The dimensionality of the merged observation is $P$ times larger than that of the original observation or equivalently, the length of the merged time-series is $P$ times shorter than that of the original one. `NCDE` is the original NCDE design in (Kidger et al., 2020). Attentive NCDE (`ANCDE`) is an extension of `NCDE` by adding an attention mechanism into it, which significantly outperforms `NCDE` (Jhin et al., 2021). For the NRDE-type baselines, we consider i) the original NRDE design (Morrill et al., 2021), and ii) the one in Fig. 1 (a) which directly embeds the higher-depth log-signature into a lower-dimensional space, each of which is called `NRDE` and `DE-NRDE`, respectively. For those NRDE-based models, we clarify its one important hyperparameter, *depth*, as part of its name, e.g., $NRDE_2$ means NRDE with $D = 2$. One important point is that `NRDE` is a generalization of `NCDE`. Therefore, $NRDE_1$ is theoretically identical to `NCDE`, if using the linear interpolation method, and for this reason, we set $D > 1$ for `NRDE`.

**Hyperparameters** When using NRDEs, there is one crucial hyperparameter, the log-signature depth $D$. We use two depth settings which were used in (Morrill et al., 2021). For our LORD-NRDE, there are two depths, $D_1$ and $D_2$, where $D_1 < D_2$. $LORD_{D_1 \to D_2}$ means that LORD-NRDE's decoder recovers the $D_2$-depth log-signature from the $D_1$-depth log-signature. The number of layers in the encoder, decoder and main NRDE, $N_g$, $N_f$, and $N_o$ of Eqs. 9 to 11, are in $\{2, 3\}$. The hidden sizes, $h_g$, $h_f$, and $h_o$ of Eqs. 9 to 11, are in $\{32, 64, 128, 192\}$. The coefficients of the $L_2$ regularizers in Eqs. 13 and 14 are in $\{1 \times 10^{-5}, 1 \times 10^{-6}\}$. The coefficient of the embedding regularizer, $c_e$ in Eq. 13 is in $\{0, 1, 10\}$. The max iteration numbers, $max\_iter_{AE}$ and $max\_iter_{TASK}$ in Alg. 1, are in $\{400, 500, 1000, 1500, 2000\}$. The learning rate of the pre-training and main-training is $1 \times 10^{-3}$. We also set $\dim(\mathbf{e}(t)) = \dim(LogSig_{r_i, r_{i+1}}^{D_1}(X))$, where $t \in [r_i, r_{i+1})$.

We also conduct experiments by setting the sub-path length $P$ to 4, 8, 32, 64, 128, 256, or 512 observations. In other words, we create one log-signature for every $P$ input observation for NCDE and NRDE-based models. For `ODE-RNN`, as described earlier, we simply concatenate $P$ observations into one observation. The final time $T$ is large in our experiments, e.g., $T > 10,000$, and the number of log-signatures is $\lfloor \frac{T}{P} \rfloor$. We test both the adjoint sensitivity method and the backpropagation through the solver.

**Evaluation Methods** We reuse the very long time-series classification and forecasting evaluation methods of (Morrill et al., 2021) and extend the methods by adding more datasets and more evaluation metrics. We use accuracy, macro F1, and ROCAUC for binary classification; accuracy, macro/weighted F1, and ROCAUC for multi-class classification; and $R^2$, explained variance, mean squared error (MSE), and mean absolute error (MAE) for forecasting — we list the complete results in Appendix D after introducing key results in the main manuscript. We also show the number of parameters for each model — for our `LORD`, we exclude the parameter numbers of the decoder since it is discarded after the pre-training. We train and test each model 5 times with different seeds. If the mean of score is the same, the smaller standard deviation is better.

## 4.2 Experimental Results

**Summary of Experimental Results** Since our main result tables have many items, we quickly summarize their highlights in Tables 2 and 3. To calculate the improvement in evaluation metrics, we use the ratio of improvement over $NRDE_2$ for each metric averaged over all the sub-path lengths and all datasets, e.g., we calculate $\frac{LORD\text{'s ROCAUC} - NRDE_2\text{'s ROCAUC}}{NRDE_2\text{'s ROCAUC}}$ and $\frac{NRDE_2\text{'s MSE} - LORD\text{'s MSE}}{NRDE_2\text{'s MSE}}$ for all classification cases and average them. The ratio of the number of parameters is calculated similarly.

Table 2: Highlights in Classification

| Method | Acc. | Mac.F1 | ROCAUC | #Params |
|---|---|---|---|---|
| ODE-RNN | -9% | -19% | -11% | -20% |
| NCDE | -4% | -8% | -5% | -64% |
| ANCDE | 0% | 0% | 0% | 15% |
| $NRDE_2$ | 0% | 0% | 0% | 0% |
| $NRDE_3$ | 2% | 4% | 2% | 537% |
| $DE-NRDE_2$ | -4% | -7% | -3% | -48% |
| $DE-NRDE_3$ | -2% | -2% | 0% | 177% |
| $LORD_{1 \to 2}$ | **16%** | **23%** | **11%** | -55% |
| $LORD_{1 \to 3}$ | 15% | 20% | 10% | -46% |
| $LORD_{2 \to 3}$ | 8% | 9% | 4% | 17% |

Table 3: Highlights in Forecasting

| Method | $R^2$ | MSE | #Params |
|---|---|---|---|
| ODE-RNN | 44% | 41% | -36% |
| NCDE | -45% | -55% | -30% |
| ANCDE | -34% | -45% | -52% |
| $NRDE_2$ | 0% | 0% | 0% |
| $NRDE_3$ | 0% | -5% | 82% |
| $DE-NRDE_2$ | 39% | 35% | -20% |
| $DE-NRDE_3$ | 62% | 59% | 31% |
| $LORD_{1 \to 2}$ | 53% | 47% | -65% |
| $LORD_{1 \to 3}$ | 51% | 46% | -59% |
| $LORD_{2 \to 3}$ | **75%** | **72%** | -39% |

Overall, LORD outperforms other baselines with smaller numbers of parameters. $LORD_{D_1 \to D_2}$ has a larger model size, excluding its decoder, than $NRDE_{D_1}$ whereas it has a much smaller model size than $NRDE_{D_2}$. However, the performance of $LORD_{D_1 \to D_2}$ is similar to or better than that of $NRDE_{D_2}$, which proves the efficacy of our method. Using the encoder-decoder structure, the high complexity of processing log-signatures can be reduced and it makes our model smaller and more amenable to train.

In many cases for LORD, the adjoint method and the backpropagation method are comparable in terms of model accuracy. Interestingly, we found that the backpropagation through the Euler method is fast enough with a neglectable sacrifice of accuracy for several cases.

**Detailed Experimental Results** Table 4 show detailed results for some selected evaluation metrics — full tables with all metrics are in Appendix D. One point is that many best outcomes are made with moderate $P$ settings. For EigenWorms, ODE-RNN can't achieve good scores for all $P$ settings. Our proposed model, LORD, achieves the best scores. In particular, $LORD_{1 \to 3}$ with $P = 32$ has a much smaller number of parameters, compared with other baselines that have similar performance. $LORD_{2 \to 3}$ also significantly outperforms $NRDE_3$ for both accuracy and mode size. For CounterMovementJump and SelfRegulationSCP2, ODE-RNN is better than other NRDE and NCDE-based baselines. However, LORD marks the best scores in all cases.

In all BIDMC experiments, LORD shows outstanding performance. $LORD_{2 \to 3}$ shows the best performance in almost all cases. Compared with NCDE and NRDE, DE-NRDE has $40 \sim 60\%$ improvements and LORD has $50 \sim 70\%$ improvements. However, LORD's model size is reduced by 60% whereas DE-NRDE's model size is reduced by 20%. Therefore, our proposed method is a better embedding method for log-signatures. Unlike EigenWorms, the time-series length in this dataset is rather short. For that reason, ODE-RNN performs better. Overall, $DE-NRDE_3$ with $P = 128$ has the best performance, among the baseline models. This means that the training difficulty by the large log-signature size can be alleviated by the direct embedding. However, LORD outperforms DE-NRDE.

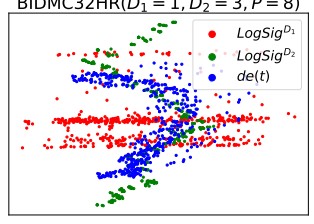

BIDMC32HR($D_1 = 1, D_2 = 3, P = 8$)

Figure 4: PCA-based visualization of log-signatures

**Additional Results and Visualization** Fig. 4 visually compares the three log-signatures and we note that the embedded signature of $d\mathbf{e}(t)$ has the characteristics of both $D_1 = 1$ and $D_2 = 3$. We refer to our supplementary material for other visualization, detailed results, sensitivity analyses, and reproducibility information.

## 5 LIMITATIONS AND CONCLUSIONS

The log-signature transform of NRDEs is suitable to irregular and long time-series. However, the log-signature transform requires larger memory footprints and to this end, we presented an autoencoder-based method to embed their higher-dimensional log-signature into a lower-dimensional space, called *LORD*. Our method is carefully devised to eradicate the higher-dimensional computation as early as right after the pre-training of the autoencoder. In the standard benchmark experiments of very long time-series, our proposed method significantly outperforms existing methods and shows relatively smaller model sizes in terms of the number of parameters. Nonetheless, it is still in its early phase to enhance NRDEs and there exist a couple of items to be studied further, e.g., processing long time-series with many channels.

Table 4: Detailed experimental results. The best results for each $P$ is in boldface and for all $P$ settings additionally with asterisk. Moderate configurations of $P$ are needed to see the best results in many cases.

| | | Accuracy | | | Macro F1 | | | #Params | | |
|---|---|---|---|---|---|---|---|---|---|---|
| | | $P = 4$ | 32 | 128 | 4 | 32 | 128 | 4 | 32 | 128 |
| EigenWorms | ODE-RNN | 0.35±0.01 | 0.35±0.03 | 0.40±0.04 | 0.11±0.00 | 0.15±0.07 | 0.25±0.11 | $2.9 \times 10^4$ | $5.4 \times 10^4$ | $1.4 \times 10^5$ |
| | NCDE | 0.68±0.10 | 0.76±0.05 | 0.46±0.05 | 0.61±0.12 | 0.69±0.07 | 0.41±0.07 | $2.1 \times 10^4$ | $2.1 \times 10^4$ | $2.1 \times 10^4$ |
| | ANCDE | 0.81±0.05 | 0.77±0.03 | 0.50±0.06 | 0.77±0.07 | 0.73±0.06 | 0.46±0.07 | $9.3 \times 10^4$ | $9.3 \times 10^4$ | $2.3 \times 10^4$ |
| | $NRDE_2$ | 0.74±0.04 | 0.78±0.03 | 0.73±0.09 | 0.66±0.08 | 0.71±0.04 | 0.66±0.11 | $6.5 \times 10^4$ | $6.5 \times 10^4$ | $6.5 \times 10^4$ |
| | $NRDE_3$ | 0.72±0.08 | 0.81±0.04 | 0.61±0.12 | 0.62±0.13 | 0.79±0.05 | 0.54±0.14 | $3.0 \times 10^5$ | $3.0 \times 10^5$ | $3.0 \times 10^5$ |
| | $DE\text{-}NRDE_2$ | 0.44±0.05 | 0.73±0.07 | 0.77±0.07 | 0.23±0.07 | 0.63±0.14 | 0.72±0.08 | $4.1 \times 10^4$ | $4.1 \times 10^4$ | $4.9 \times 10^4$ |
| | $DE\text{-}NRDE_3$ | 0.42±0.07 | 0.63±0.05 | **0.84±0.04** | 0.21±0.10 | 0.53±0.04 | 0.81±0.03 | $1.8 \times 10^5$ | $1.8 \times 10^5$ | $2.4 \times 10^5$ |
| | $LORD_{1 \to 2}$ | 0.80±0.05 | 0.84±0.05 | 0.76±0.06 | 0.74±0.05 | 0.84±0.06 | 0.75±0.08 | $3.9 \times 10^4$ | $3.9 \times 10^4$ | $3.5 \times 10^4$ |
| | $LORD_{1 \to 3}$ | 0.77±0.09 | **0.86±0.05***​ | 0.74±0.08 | 0.73±0.12 | **0.85±0.06***​ | 0.72±0.09 | $3.5 \times 10^4$ | $3.5 \times 10^4$ | $3.9 \times 10^4$ |
| | $LORD_{2 \to 3}$ | **0.82±0.05** | 0.84±0.02 | **0.84±0.04** | **0.78±0.06** | 0.79±0.05 | **0.82±0.06** | $1.3 \times 10^5$ | $1.3 \times 10^5$ | $1.3 \times 10^5$ |
| | | $P = 64$ | 128 | 512 | 64 | 128 | 512 | 64 | 128 | 512 |
| CounterMovementJump | ODE-RNN | 0.47±0.02 | 0.47±0.09 | 0.45±0.03 | 0.45±0.03 | 0.46±0.09 | 0.44±0.04 | $2.5 \times 10^4$ | $2.1 \times 10^5$ | $1.4 \times 10^5$ |
| | NCDE | 0.35±0.06 | 0.40±0.06 | 0.41±0.05 | 0.32±0.09 | 0.35±0.05 | 0.38±0.06 | $5.8 \times 10^4$ | $2.5 \times 10^4$ | $4.7 \times 10^4$ |
| | ANCDE | 0.35±0.04 | 0.47±0.06 | 0.46±0.07 | 0.34±0.03 | 0.46±0.07 | 0.43±0.10 | $1.4 \times 10^5$ | $2.5 \times 10^5$ | $2.9 \times 10^5$ |
| | $NRDE_2$ | 0.40±0.12 | 0.40±0.08 | 0.40±0.09 | 0.39±0.12 | 0.38±0.08 | 0.39±0.09 | $1.1 \times 10^5$ | $1.9 \times 10^5$ | $5.0 \times 10^4$ |
| | $NRDE_3$ | 0.47±0.09 | 0.41±0.08 | 0.47±0.06 | 0.46±0.09 | 0.40±0.08 | 0.45±0.05 | $5.3 \times 10^5$ | $5.2 \times 10^5$ | $2.1 \times 10^6$ |
| | $DE\text{-}NRDE_2$ | 0.40±0.07 | 0.38±0.06 | 0.38±0.05 | 0.39±0.08 | 0.33±0.07 | 0.36±0.06 | $5.2 \times 10^4$ | $9.1 \times 10^4$ | $3.9 \times 10^4$ |
| | $DE\text{-}NRDE_3$ | 0.41±0.07 | 0.47±0.04 | 0.42±0.07 | 0.37±0.07 | 0.44±0.05 | 0.40±0.08 | $2.0 \times 10^5$ | $2.8 \times 10^5$ | $7.3 \times 10^5$ |
| | $LORD_{1 \to 2}$ | **0.59±0.04***​ | 0.57±0.06 | 0.50±0.08 | **0.58±0.04***​ | 0.56±0.07 | 0.49±0.08 | $8.7 \times 10^4$ | $5.1 \times 10^4$ | $6.5 \times 10^4$ |
| | $LORD_{1 \to 3}$ | 0.56±0.04 | 0.53±0.03 | **0.51±0.03** | 0.55±0.04 | 0.52±0.03 | **0.50±0.03** | $4.3 \times 10^4$ | $1.4 \times 10^5$ | $8.8 \times 10^4$ |
| | $LORD_{2 \to 3}$ | 0.40±0.05 | 0.40±0.08 | 0.40±0.04 | 0.38±0.05 | 0.38±0.08 | 0.37±0.02 | $1.1 \times 10^5$ | $1.5 \times 10^5$ | $3.6 \times 10^4$ |
| | | $P = 32$ | 64 | 256 | 32 | 64 | 256 | 32 | 64 | 256 |
| SelfRegulationSCP2 | ODE-RNN | 0.55±0.05 | 0.58±0.01 | 0.56±0.04 | 0.54±0.06 | 0.46±0.05 | 0.48±0.04 | $5.7 \times 10^4$ | $4.1 \times 10^4$ | $1.4 \times 10^5$ |
| | NCDE | 0.59±0.07 | 0.52±0.08 | 0.54±0.06 | 0.47±0.12 | 0.46±0.16 | 0.54±0.09 | $4.2 \times 10^4$ | $2.2 \times 10^5$ | $3.2 \times 10^5$ |
| | ANCDE | 0.60±0.06 | 0.54±0.04 | 0.49±0.05 | 0.55±0.05 | 0.49±0.05 | 0.43±0.07 | $3.9 \times 10^5$ | $1.3 \times 10^5$ | $2.1 \times 10^5$ |
| | $NRDE_2$ | 0.52±0.11 | 0.56±0.09 | 0.52±0.11 | 0.51±0.06 | 0.45±0.06 | 0.50±0.07 | $3.2 \times 10^5$ | $6.2 \times 10^5$ | $6.2 \times 10^5$ |
| | $NRDE_3$ | 0.54±0.05 | 0.56±0.06 | 0.50±0.07 | 0.56±0.05 | 0.50±0.12 | 0.48±0.07 | $3.4 \times 10^6$ | $1.7 \times 10^6$ | $3.4 \times 10^6$ |
| | $DE\text{-}NRDE_2$ | 0.59±0.10 | 0.55±0.06 | 0.53±0.08 | 0.53±0.13 | 0.50±0.08 | 0.50±0.08 | $1.1 \times 10^5$ | $2.0 \times 10^5$ | $2.0 \times 10^5$ |
| | $DE\text{-}NRDE_3$ | 0.55±0.11 | 0.51±0.08 | 0.51±0.10 | 0.57±0.07 | 0.54±0.06 | 0.55±0.03 | $1.1 \times 10^6$ | $5.4 \times 10^5$ | $1.1 \times 10^6$ |
| | $LORD_{1 \to 2}$ | 0.57±0.10 | 0.58±0.12 | 0.53±0.08 | **0.57±0.06** | **0.61±0.06***​ | 0.51±0.08 | $2.4 \times 10^4$ | $1.4 \times 10^5$ | $7.5 \times 10^4$ |
| | $LORD_{1 \to 3}$ | 0.57±0.06 | **0.62±0.10***​ | 0.53±0.10 | 0.50±0.06 | 0.59±0.11 | **0.56±0.04** | $1.4 \times 10^5$ | $1.2 \times 10^5$ | $1.6 \times 10^5$ |
| | $LORD_{2 \to 3}$ | **0.61±0.06** | 0.61±0.10 | **0.60±0.07** | 0.55±0.09 | 0.56±0.07 | 0.53±0.06 | $2.2 \times 10^5$ | $5.3 \times 10^5$ | $2.6 \times 10^5$ |

| | | $R^2$ | | | MSE | | | #Params | | |
|---|---|---|---|---|---|---|---|---|---|---|
| | | $P = 8$ | 128 | 512 | 8 | 128 | 512 | 8 | 128 | 512 |
| BIDMC32HR | ODE-RNN | 0.57±0.29 | 0.92±0.01 | 0.81±0.02 | 0.41±0.28 | 0.07±0.01 | 0.18±0.02 | $4 \times 10^3$ | $1.5 \times 10^4$ | $5.2 \times 10^4$ |
| | NCDE | 0.39±0.04 | 0.23±0.04 | 0.19±0.04 | 0.59±0.04 | 0.74±0.04 | 0.78±0.04 | $5.0 \times 10^4$ | $5.0 \times 10^4$ | $5.0 \times 10^4$ |
| | ANCDE | 0.44±0.03 | 0.17±0.06 | 0.26±0.01 | 0.54±0.03 | 0.80±0.06 | 0.71±0.01 | $4.7 \times 10^4$ | $4.7 \times 10^4$ | $4.7 \times 10^4$ |
| | $NRDE_2$ | 0.63±0.04 | 0.67±0.07 | 0.64±0.07 | 0.36±0.04 | 0.32±0.07 | 0.34±0.07 | $7.5 \times 10^4$ | $7.5 \times 10^4$ | $7.5 \times 10^4$ |
| | $NRDE_3$ | 0.65±0.08 | 0.58±0.13 | 0.40±0.13 | 0.34±0.08 | 0.40±0.13 | 0.58±0.13 | $1.4 \times 10^5$ | $1.4 \times 10^5$ | $1.4 \times 10^5$ |
| | $DE\text{-}NRDE_2$ | 0.82±0.08 | 0.76±0.05 | 0.64±0.02 | 0.17±0.07 | 0.23±0.05 | 0.35±0.02 | $6.3 \times 10^4$ | $6.0 \times 10^4$ | $5.9 \times 10^4$ |
| | $DE\text{-}NRDE_3$ | 0.89±0.05 | 0.93±0.02 | 0.81±0.09 | 0.11±0.05 | 0.07±0.02 | 0.18±0.08 | $1.2 \times 10^5$ | $1.0 \times 10^5$ | $1.0 \times 10^5$ |
| | $LORD_{1 \to 2}$ | **0.98±0.01***​ | 0.94±0.01 | 0.44±0.04 | **0.02±0.01***​ | 0.06±0.01 | 0.54±0.04 | $2.7 \times 10^4$ | $3.4 \times 10^4$ | $3.4 \times 10^4$ |
| | $LORD_{1 \to 3}$ | **0.98±0.01***​ | 0.92±0.01 | 0.45±0.03 | **0.02±0.01***​ | 0.07±0.01 | 0.53±0.03 | $2.7 \times 10^4$ | $6.5 \times 10^4$ | $5.5 \times 10^4$ |
| | $LORD_{2 \to 3}$ | **0.98±0.01***​ | **0.95±0.01** | **0.85±0.04** | **0.02±0.01***​ | **0.04±0.01** | **0.15±0.04** | $3.6 \times 10^4$ | $5.2 \times 10^4$ | $5.4 \times 10^4$ |
| BIDMC32RR | ODE-RNN | 0.54±0.20 | 0.72±0.01 | 0.65±0.02 | 0.45±0.19 | 0.27±0.01 | 0.34±0.02 | $5.4 \times 10^4$ | $1.2 \times 10^5$ | $3.4 \times 10^5$ |
| | NCDE | 0.21±0.03 | 0.34±0.05 | 0.24±0.03 | 0.76±0.03 | 0.64±0.05 | 0.74±0.03 | $8.7 \times 10^4$ | $8.7 \times 10^4$ | $8.7 \times 10^4$ |
| | ANCDE | 0.30±0.06 | 0.39±0.10 | 0.27±0.03 | 0.68±0.06 | 0.59±0.09 | 0.70±0.03 | $4.7 \times 10^4$ | $4.7 \times 10^4$ | $5.6 \times 10^4$ |
| | $NRDE_2$ | 0.27±0.04 | 0.52±0.08 | 0.50±0.16 | 0.70±0.04 | 0.46±0.08 | 0.48±0.16 | $1.2 \times 10^5$ | $1.2 \times 10^5$ | $1.2 \times 10^5$ |
| | $NRDE_3$ | 0.34±0.04 | 0.65±0.14 | 0.42±0.13 | 0.64±0.03 | 0.34±0.13 | 0.56±0.13 | $2.2 \times 10^5$ | $2.2 \times 10^5$ | $2.2 \times 10^5$ |
| | $DE\text{-}NRDE_2$ | 0.64±0.01 | 0.58±0.06 | 0.55±0.06 | 0.34±0.01 | 0.40±0.06 | 0.43±0.06 | $8.8 \times 10^4$ | $1.0 \times 10^5$ | $1.0 \times 10^5$ |
| | $DE\text{-}NRDE_3$ | 0.70±0.06 | 0.70±0.02 | 0.57±0.05 | 0.29±0.05 | 0.29±0.02 | 0.41±0.05 | $1.4 \times 10^5$ | $1.6 \times 10^5$ | $1.6 \times 10^5$ |
| | $LORD_{1 \to 2}$ | **0.81±0.02***​ | 0.66±0.01 | 0.45±0.04 | **0.18±0.02***​ | 0.33±0.01 | 0.53±0.04 | $2.7 \times 10^4$ | $2.7 \times 10^4$ | $4.0 \times 10^4$ |
| | $LORD_{1 \to 3}$ | 0.80±0.02 | 0.67±0.02 | 0.45±0.03 | 0.20±0.02 | 0.32±0.01 | 0.53±0.03 | $2.7 \times 10^4$ | $4.5 \times 10^4$ | $4.5 \times 10^4$ |
| | $LORD_{2 \to 3}$ | 0.80±0.01 | **0.74±0.01** | **0.66±0.01** | 0.19±0.01 | **0.25±0.01** | **0.33±0.01** | $3.6 \times 10^4$ | $1.4 \times 10^5$ | $1.0 \times 10^5$ |
| BIDMC32SpO2 | ODE-RNN | 0.55±0.11 | 0.90±0.03 | 0.75±0.02 | 0.46±0.11 | 0.10±0.03 | 0.26±0.03 | $4 \times 10^3$ | $1.5 \times 10^4$ | $5.2 \times 10^4$ |
| | NCDE | 0.24±0.07 | 0.21±0.07 | 0.33±0.04 | 0.79±0.07 | 0.81±0.08 | 0.69±0.05 | $8.7 \times 10^4$ | $8.7 \times 10^4$ | $8.7 \times 10^4$ |
| | ANCDE | 0.31±0.03 | 0.33±0.05 | 0.37±0.03 | 0.71±0.03 | 0.70±0.05 | 0.65±0.03 | $4.7 \times 10^4$ | $4.7 \times 10^4$ | $4.7 \times 10^4$ |
| | $NRDE_2$ | 0.22±0.08 | 0.47±0.10 | 0.61±0.18 | 0.81±0.08 | 0.55±0.10 | 0.41±0.19 | $1.2 \times 10^5$ | $1.2 \times 10^5$ | $1.2 \times 10^5$ |
| | $NRDE_3$ | 0.13±0.09 | 0.68±0.15 | 0.60±0.18 | 0.90±0.09 | 0.33±0.15 | 0.42±0.19 | $2.2 \times 10^5$ | $2.2 \times 10^5$ | $2.2 \times 10^5$ |
| | $DE\text{-}NRDE_2$ | 0.85±0.01 | 0.74±0.05 | 0.59±0.04 | 0.15±0.01 | 0.27±0.05 | 0.43±0.04 | $8.8 \times 10^4$ | $1.0 \times 10^5$ | $1.0 \times 10^5$ |
| | $DE\text{-}NRDE_3$ | 0.90±0.08 | 0.93±0.03 | 0.76±0.07 | 0.10±0.08 | 0.08±0.03 | 0.25±0.07 | $1.4 \times 10^5$ | $1.6 \times 10^5$ | $1.6 \times 10^5$ |
| | $LORD_{1 \to 2}$ | 0.97±0.00 | 0.91±0.01 | 0.58±0.05 | 0.03±0.00 | 0.09±0.01 | 0.43±0.05 | $2.7 \times 10^4$ | $3.4 \times 10^4$ | $6.2 \times 10^4$ |
| | $LORD_{1 \to 3}$ | **0.98±0.01***​ | 0.89±0.03 | 0.52±0.04 | **0.02±0.01***​ | 0.12±0.03 | 0.49±0.05 | $2.7 \times 10^4$ | $2.7 \times 10^4$ | $3.8 \times 10^4$ |
| | $LORD_{2 \to 3}$ | **0.98±0.01***​ | **0.94±0.01** | **0.83±0.03** | **0.02±0.01***​ | **0.06±0.01** | **0.18±0.03** | $3.6 \times 10^4$ | $5.6 \times 10^4$ | $6.9 \times 10^4$ |

## 6 ACKNOWLEDGEMENTS

Noseong Park is the corresponding author. This work was supported by the Yonsei University Research Fund of 2021, and the Institute of Information & Communications Technology Planning & Evaluation (IITP) grant funded by the Korean government (MSIT) (No. 2020-0-01361, Artificial Intelligence Graduate School Program (Yonsei University), and No. 2021-0-00155, Context and Activity Analysis-based Solution for Safe Childcare).

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

Table 5: The best hyperparameter in `ODE-RNN`

| Data | $P$ | Hidden Path Size |
|---|---|---|
| Counter-MovementJump | 64 | 64 |
| | 128 | 256 |
| | 512 | 64 |
| Self-RegulationSCP2 | 32 | 128 |
| | 64 | 64 |
| | 256 | 64 |

Table 6: The best hyperparameter in `NCDE` and `NRDE`

| Data | $P$ | Hidden Path Size | | | CDE Function's Hidden Size | | | #Hidden Layers | | |
|---|---|---|---|---|---|---|---|---|---|---|
| | | $D=1$ | 2 | 3 | 1 | 2 | 3 | 1 | 2 | 3 |
| Counter-MovementJump | 64 | 64 | 64 | 128 | 128 | 128 | 128 | 3 | 2 | 3 |
| | 128 | 64 | 256 | 256 | 64 | 64 | 64 | 3 | 2 | 3 |
| | 512 | 128 | 64 | 256 | 64 | 64 | 256 | 3 | 2 | 3 |
| Self-RegulationSCP2 | 32 | 64 | 64 | 128 | 64 | 128 | 128 | 2 | 2 | 2 |
| | 64 | 64 | 256 | 128 | 256 | 64 | 64 | 3 | 2 | 2 |
| | 256 | 256 | 256 | 64 | 128 | 64 | 256 | 3 | 2 | 3 |

Table 7: The best hyperparameter in `ANCDE`

| Data | $P$ | Hidden Path Size | #Hidden Layers | Attention Channel Size |
|---|---|---|---|---|
| EigenWorms | 4 | 128 | 3 | 64 |
| | 32 | 128 | 3 | 64 |
| | 128 | 64 | 2 | 32 |
| Counter-MovementJump | 64 | 128 | 2 | 128 |
| | 128 | 256 | 3 | 128 |
| | 512 | 64 | 2 | 256 |
| Self-RegulationSCP2 | 32 | 256 | 3 | 128 |
| | 64 | 64 | 2 | 128 |
| | 256 | 128 | 2 | 128 |
| BIDMC32HR | 8 | 128 | 2 | 64 |
| | 128 | 128 | 2 | 64 |
| | 512 | 128 | 2 | 64 |
| BIDMC32RR | 8 | 128 | 2 | 64 |
| | 128 | 128 | 2 | 64 |
| | 512 | 128 | 3 | 64 |
| BIDMC32SpO2 | 8 | 128 | 2 | 64 |
| | 128 | 128 | 2 | 64 |
| | 512 | 128 | 2 | 64 |

Table 8: The best hyperparameter in `DE-NRDE`

| Data | $P$ | Compression Ratio | | #Hidden Layers | | Hidden Size | |
|---|---|---|---|---|---|---|---|
| | | $D=2$ | 3 | 2 | 3 | 2 | 3 |
| EigenWorms | 4 | 0.5 | 0.5 | 1 | 1 | 128 | 128 |
| | 32 | 0.5 | 0.5 | 1 | 1 | 128 | 128 |
| | 128 | 0.7 | 0.7 | 1 | 1 | 64 | 128 |
| Counter-MovementJump | 64 | 0.3 | 0.3 | 1 | 2 | 128 | 128 |
| | 128 | 0.3 | 0.5 | 2 | 2 | 128 | 64 |
| | 512 | 0.7 | 0.3 | 1 | 1 | 64 | 128 |
| Self-RegulationSCP2 | 32 | 0.3 | 0.3 | 1 | 2 | 64 | 128 |
| | 64 | 0.3 | 0.3 | 2 | 2 | 64 | 64 |
| | 256 | 0.3 | 0.3 | 2 | 2 | 64 | 128 |
| BIDMC32HR | 8 | 0.7 | 0.7 | 2 | 2 | 64 | 128 |
| | 128 | 0.7 | 0.7 | 1 | 1 | 128 | 128 |
| | 512 | 0.7 | 0.7 | 1 | 1 | 64 | 128 |
| BIDMC32RR | 8 | 0.5 | 0.5 | 1 | 1 | 128 | 128 |
| | 128 | 0.7 | 0.7 | 1 | 1 | 128 | 128 |
| | 512 | 0.7 | 0.7 | 1 | 1 | 64 | 64 |
| BIDMC32SpO2 | 8 | 0.5 | 0.5 | 1 | 1 | 128 | 128 |
| | 128 | 0.7 | 0.7 | 1 | 1 | 64 | 64 |
| | 512 | 0.7 | 0.7 | 1 | 1 | 64 | 64 |

## A  BEST HYPERPARAMETERS

Our software and hardware environments are as follows: UBUNTU 18.04 LTS, PYTHON 3.7.10, PYTORCH 1.8.1, CUDA 11.4, and NVIDIA Driver 470.42.01, i9 CPU, and NVIDIA RTX A6000.

### A.1  BASELINE

There are 5 baselines in our experiments. Each model has its own hyerparameters. `ODE-RNN` method has a hyperparameter of hidden path size. `NCDE` and `NRDE` have also the same hyperparameter, and in addition, the size of hidden dimension and the number of hidden layers in their CDE and RDE functions. The Attentive NCDE (`ANCDE`) has all of those hyperparameters and in addition, the attention channel size. In `DE-NRDE`, there are addition hyperparameters related to embedding, which are the embedding compression ratio, the size of hidden dimension, and the number of hidden layers in the embedding layer.

In (Morrill et al., 2021), there are three models ,`ODE-RNN`, `NCDE` and `NRDE`, and four datasets, `EigenWorms`, `BIDMC32HR`, `BIDMC32RR`, and `BIDMC32SpO2`. For these combinations, we set the hyperparameters as reported in (Morrill et al., 2021). In other cases, we find the best hyperparameters in the following range. All kinds of hidden size in `ODE-RNN`, `NCDE`, `ANCDE`, and `NRDE` are in {32, 64, 128, 256}. The number of hidden layers in those is in {2,3}. In `DE-NRDE`, the embedding compression ratio, the hidden size, and the number of hidden layers in the embedding layer are in {0.3, 0.5, 0.7}, {64, 128}, and {1,2}, respectively. Tables 5 to 8 show the best hyperparameters of the baselines.

### A.2  LORD-NRDE

Table 9 shows the best hyperparameter configuration of our method in each dataset.

Table 9: The best hyperparameter in `LORD`

| Method | Data | $P$ | $N_f$ | $N_g$ | $N_o$ | $h_f$ | $h_g$ | $h_o$ | $c_{AE}$ | $c_{TASK}$ | $c_e$ | max_iter AE | max_iter TASK |
|---|---|---|---|---|---|---|---|---|---|---|---|---|---|
| $LORD_{1\to2}$ | EigenWorms | 4 | 3 | 3 | 3 | 64 | 64 | 64 | $1\times10^{-6}$ | $1\times10^{-6}$ | 0 | 400 | 400 |
| | | 32 | 3 | 3 | 3 | 64 | 64 | 64 | $1\times10^{-6}$ | $1\times10^{-6}$ | 0 | 2000 | 400 |
| | | 128 | 3 | 2 | 3 | 64 | 64 | 64 | $1\times10^{-6}$ | $1\times10^{-6}$ | 0 | 2000 | 400 |
| | Counter-MovementJump | 64 | 2 | 3 | 2 | 32 | 128 | 32 | $1\times10^{-6}$ | $1\times10^{-6}$ | 1 | 500 | 400 |
| | | 128 | 3 | 3 | 3 | 64 | 64 | 64 | $1\times10^{-6}$ | $1\times10^{-6}$ | 1 | 500 | 400 |
| | | 512 | 2 | 2 | 2 | 64 | 128 | 64 | $1\times10^{-6}$ | $1\times10^{-6}$ | 1 | 1500 | 400 |
| | Self-RegulationSCP2 | 32 | 3 | 3 | 3 | 32 | 32 | 32 | $1\times10^{-6}$ | $1\times10^{-6}$ | 1 | 1000 | 400 |
| | | 64 | 3 | 2 | 3 | 128 | 128 | 128 | $1\times10^{-6}$ | $1\times10^{-6}$ | 1 | 500 | 400 |
| | | 256 | 3 | 3 | 3 | 128 | 32 | 128 | $1\times10^{-6}$ | $1\times10^{-6}$ | 1 | 500 | 400 |
| | BIDMC32HR | 8 | 3 | 3 | 3 | 64 | 64 | 64 | $1\times10^{-5}$ | $1\times10^{-5}$ | 0 | 400 | 2000 |
| | | 128 | 3 | 3 | 3 | 32 | 64 | 32 | $1\times10^{-5}$ | $1\times10^{-5}$ | 0 | 1000 | 500 |
| | | 512 | 3 | 3 | 3 | 32 | 64 | 32 | $1\times10^{-5}$ | $1\times10^{-5}$ | 0 | 1000 | 500 |
| | BIDMC32RR | 8 | 3 | 3 | 3 | 64 | 64 | 64 | $1\times10^{-5}$ | $1\times10^{-5}$ | 0 | 400 | 2000 |
| | | 128 | 3 | 3 | 3 | 64 | 64 | 64 | $1\times10^{-5}$ | $1\times10^{-5}$ | 0 | 1000 | 500 |
| | | 512 | 3 | 3 | 3 | 64 | 64 | 64 | $1\times10^{-5}$ | $1\times10^{-5}$ | 0 | 1000 | 500 |
| | BIDMC32SpO2 | 8 | 3 | 3 | 3 | 64 | 64 | 64 | $1\times10^{-5}$ | $1\times10^{-5}$ | 0 | 400 | 2000 |
| | | 128 | 3 | 3 | 3 | 32 | 64 | 32 | $1\times10^{-5}$ | $1\times10^{-5}$ | 0 | 1000 | 500 |
| | | 512 | 3 | 3 | 3 | 64 | 64 | 64 | $1\times10^{-5}$ | $1\times10^{-5}$ | 0 | 1000 | 500 |
| $LORD_{1\to3}$ | EigenWorms | 4 | 3 | 2 | 3 | 64 | 64 | 64 | $1\times10^{-6}$ | $1\times10^{-6}$ | 0 | 400 | 400 |
| | | 32 | 3 | 2 | 3 | 64 | 64 | 64 | $1\times10^{-6}$ | $1\times10^{-6}$ | 0 | 2000 | 400 |
| | | 128 | 3 | 3 | 3 | 64 | 64 | 64 | $1\times10^{-6}$ | $1\times10^{-6}$ | 0 | 2000 | 400 |
| | Counter-MovementJump | 64 | 2 | 3 | 2 | 32 | 64 | 32 | $1\times10^{-6}$ | $1\times10^{-6}$ | 1 | 500 | 400 |
| | | 128 | 3 | 3 | 3 | 128 | 128 | 128 | $1\times10^{-6}$ | $1\times10^{-6}$ | 1 | 500 | 400 |
| | | 512 | 3 | 2 | 3 | 32 | 128 | 32 | $1\times10^{-6}$ | $1\times10^{-6}$ | 1 | 500 | 400 |
| | Self-RegulationSCP2 | 32 | 2 | 3 | 2 | 32 | 128 | 32 | $1\times10^{-6}$ | $1\times10^{-6}$ | 1 | 500 | 400 |
| | | 64 | 3 | 3 | 3 | 32 | 128 | 32 | $1\times10^{-6}$ | $1\times10^{-6}$ | 1 | 1500 | 400 |
| | | 256 | 3 | 3 | 3 | 128 | 128 | 128 | $1\times10^{-6}$ | $1\times10^{-6}$ | 1 | 1500 | 400 |
| | BIDMC32HR | 8 | 3 | 3 | 3 | 128 | 64 | 128 | $1\times10^{-5}$ | $1\times10^{-5}$ | 0 | 400 | 2000 |
| | | 128 | 3 | 3 | 3 | 32 | 64 | 32 | $1\times10^{-5}$ | $1\times10^{-5}$ | 0 | 1000 | 500 |
| | | 512 | 3 | 3 | 3 | 32 | 64 | 32 | $1\times10^{-5}$ | $1\times10^{-5}$ | 0 | 1000 | 500 |
| | BIDMC32RR | 8 | 3 | 3 | 3 | 64 | 64 | 64 | $1\times10^{-5}$ | $1\times10^{-5}$ | 0 | 400 | 2000 |
| | | 128 | 3 | 3 | 3 | 64 | 64 | 64 | $1\times10^{-5}$ | $1\times10^{-5}$ | 0 | 1000 | 500 |
| | | 512 | 3 | 3 | 3 | 64 | 64 | 64 | $1\times10^{-5}$ | $1\times10^{-5}$ | 0 | 1000 | 500 |
| | BIDMC32SpO2 | 8 | 3 | 3 | 3 | 64 | 64 | 64 | $1\times10^{-5}$ | $1\times10^{-5}$ | 0 | 400 | 2000 |
| | | 128 | 3 | 3 | 3 | 64 | 64 | 64 | $1\times10^{-5}$ | $1\times10^{-5}$ | 0 | 1000 | 500 |
| | | 512 | 3 | 3 | 3 | 32 | 64 | 32 | $1\times10^{-5}$ | $1\times10^{-5}$ | 0 | 1000 | 500 |
| $LORD_{2\to3}$ | EigenWorms | 4 | 3 | 3 | 3 | 64 | 64 | 64 | $1\times10^{-6}$ | $1\times10^{-6}$ | 0 | 400 | 400 |
| | | 32 | 3 | 3 | 3 | 64 | 64 | 64 | $1\times10^{-6}$ | $1\times10^{-6}$ | 0 | 2000 | 400 |
| | | 128 | 3 | 2 | 3 | 64 | 64 | 64 | $1\times10^{-6}$ | $1\times10^{-6}$ | 0 | 2000 | 400 |
| | Counter-MovementJump | 64 | 3 | 2 | 3 | 128 | 128 | 128 | $1\times10^{-6}$ | $1\times10^{-6}$ | 1 | 1500 | 400 |
| | | 128 | 2 | 2 | 2 | 64 | 128 | 64 | $1\times10^{-6}$ | $1\times10^{-6}$ | 1 | 1000 | 400 |
| | | 512 | 3 | 2 | 3 | 64 | 32 | 64 | $1\times10^{-6}$ | $1\times10^{-6}$ | 1 | 1000 | 400 |
| | Self-RegulationSCP2 | 32 | 3 | 2 | 3 | 32 | 64 | 32 | $1\times10^{-6}$ | $1\times10^{-6}$ | 1 | 500 | 400 |
| | | 64 | 2 | 3 | 2 | 128 | 128 | 128 | $1\times10^{-6}$ | $1\times10^{-6}$ | 1 | 1000 | 400 |
| | | 256 | 2 | 3 | 2 | 32 | 128 | 32 | $1\times10^{-6}$ | $1\times10^{-6}$ | 1 | 500 | 400 |
| | BIDMC32HR | 8 | 3 | 3 | 3 | 64 | 64 | 64 | $1\times10^{-5}$ | $1\times10^{-5}$ | 0 | 400 | 2000 |
| | | 128 | 3 | 3 | 3 | 32 | 64 | 32 | $1\times10^{-5}$ | $1\times10^{-5}$ | 0 | 1000 | 500 |
| | | 512 | 3 | 3 | 3 | 64 | 64 | 64 | $1\times10^{-5}$ | $1\times10^{-5}$ | 0 | 1000 | 500 |
| | BIDMC32RR | 8 | 3 | 3 | 3 | 64 | 64 | 64 | $1\times10^{-5}$ | $1\times10^{-5}$ | 0 | 400 | 2000 |
| | | 128 | 3 | 3 | 3 | 64 | 192 | 64 | $1\times10^{-5}$ | $1\times10^{-5}$ | 0 | 1000 | 500 |
| | | 512 | 3 | 3 | 3 | 128 | 64 | 128 | $1\times10^{-5}$ | $1\times10^{-5}$ | 0 | 1000 | 500 |
| | BIDMC32SpO2 | 8 | 3 | 3 | 3 | 64 | 64 | 64 | $1\times10^{-5}$ | $1\times10^{-5}$ | 0 | 400 | 2000 |
| | | 128 | 3 | 3 | 3 | 64 | 64 | 64 | $1\times10^{-5}$ | $1\times10^{-5}$ | 0 | 1000 | 500 |
| | | 512 | 3 | 3 | 3 | 128 | 64 | 128 | $1\times10^{-5}$ | $1\times10^{-5}$ | 0 | 1000 | 500 |

# B   DATASET

`EigenWorms` has a time-series length of 17,984 and a channel of 7 which contains the movement data of roundworms. The goal is to classify each worm among 5 worm types. `CounterMovementJump` has 4,250 length and 4 channels which means accelerations data of each 3D-axis. Using accelerations data, the type of jump is predicted among 3 types. The object of `SelfRegulationSCP2` is to classify whether the subject moves the computer's cursor up or down, using 8 channels of the EEG data. Its time-series length is 1,153.

For forecasting, three Beth Israel Deaconess Medical Centre (`BIDMC`) datasets are used. Using the PPG and ECG information, each task is to predict a person's heart rate (`HR`), respiratory rate (`RR`), or oxygen saturation (`SpO2`), respectively. The time-series length is 4,000.

## C  VISUALIZATION

We show other PCA based visualizations of the log-signatures. Using PCA, we extract the distributions of the most important principle components of $\{LogSig_{r_i,r_{i+1}}^{D_1}(X)\}_{i=0}^{\lfloor \frac{T}{P} \rfloor - 1}$, $\{LogSig_{r_i,r_{i+1}}^{D_2}(X)\}_{i=0}^{\lfloor \frac{T}{P} \rfloor - 1}$, and $d\mathbf{e}(t)$. If two paths have similar distributions on those components, the information contained by the two paths is similar. In Fig. 5, the distribution of $d\mathbf{e}(t)$ is somewhere in between the distributions of $\{LogSig_{r_i,r_{i+1}}^{D_2}(X)\}_{i=0}^{\lfloor \frac{T}{P} \rfloor - 1}$ and $\{LogSig_{r_i,r_{i+1}}^{D_1}(X)\}_{i=0}^{\lfloor \frac{T}{P} \rfloor - 1}$. From the encoder-decoder learning, $d\mathbf{e}(t)$ can learn the information of $\{LogSig_{r_i,r_{i+1}}^{D_2}(X)\}_{i=0}^{\lfloor \frac{T}{P} \rfloor - 1}$.

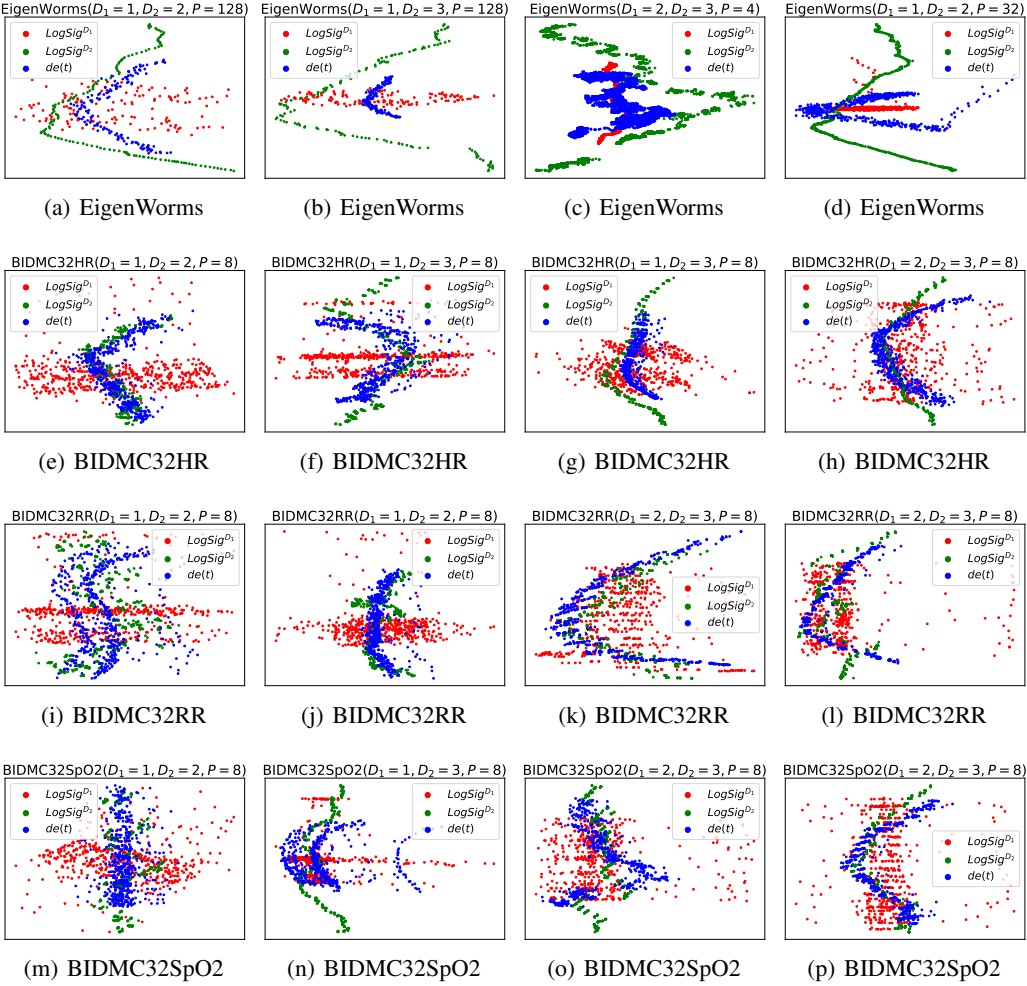

Figure 5: Visualization of $\{LogSig_{r_i,r_{i+1}}^{D_1}(X)\}_{i=0}^{\lfloor \frac{T}{P} \rfloor - 1}$, $\{LogSig_{r_i,r_{i+1}}^{D_2}(X)\}_{i=0}^{\lfloor \frac{T}{P} \rfloor - 1}$, and $d\mathbf{e}(t)$ in `EigenWorms` and `BIDMC`.

Table 10: `EigenWorms`

| Method | $P$ | Accuracy | Macro F1 | Weighted F1 | ROCAUC | #Params(D) | #Params(R) |
|---|---|---|---|---|---|---|---|
| ODE-RNN | 4 | 0.354±0.011 | 0.107±0.002 | 0.193±0.004 | 0.511±0.054 | Not Applicable | 28707 |
| | 32 | 0.349±0.029 | 0.150±0.065 | 0.232±0.064 | 0.439±0.027 | Not Applicable | 53795 |
| | 128 | 0.395±0.043 | 0.249±0.110 | 0.304±0.095 | 0.517±0.068 | Not Applicable | 139811 |
| NCDE | 4 | 0.677±0.099 | 0.610±0.122 | 0.665±0.105 | 0.913±0.018 | Not Applicable | 21253 |
| | 32 | 0.759±0.047 | 0.690±0.065 | 0.756±0.052 | 0.907±0.016 | Not Applicable | 21253 |
| | 128 | 0.456±0.046 | 0.415±0.066 | 0.458±0.051 | 0.678±0.055 | Not Applicable | 21253 |
| ANCDE | 4 | 0.815±0.046 | 0.770±0.065 | 0.810±0.046 | 0.957±0.012 | Not Applicable | 92726 |
| | 32 | 0.774±0.033 | 0.727±0.060 | 0.770±0.033 | 0.925±0.021 | Not Applicable | 92726 |
| | 128 | 0.497±0.064 | 0.456±0.074 | 0.501±0.063 | 0.705±0.038 | Not Applicable | 22806 |
| NRDE$_2$ | 4 | 0.744±0.044 | 0.655±0.076 | 0.725±0.059 | 0.915±0.032 | Not Applicable | 64933 |
| | 32 | 0.779±0.029 | 0.713±0.044 | 0.774±0.027 | 0.934±0.011 | Not Applicable | 64933 |
| | 128 | 0.728±0.088 | 0.661±0.106 | 0.726±0.092 | 0.908±0.037 | Not Applicable | 64933 |
| NRDE$_3$ | 4 | 0.718±0.083 | 0.616±0.127 | 0.695±0.101 | 0.914±0.023 | Not Applicable | 297893 |
| | 32 | 0.810±0.043 | 0.787±0.051 | 0.814±0.044 | 0.911±0.020 | Not Applicable | 297893 |
| | 128 | 0.610±0.117 | 0.544±0.142 | 0.600±0.120 | 0.862±0.063 | Not Applicable | 297893 |
| DE-NRDE$_2$ | 4 | 0.436±0.048 | 0.230±0.073 | 0.303±0.066 | 0.538±0.045 | Not Applicable | 41331 |
| | 32 | 0.728±0.074 | 0.631±0.141 | 0.711±0.095 | 0.918±0.026 | Not Applicable | 41331 |
| | 128 | 0.769±0.068 | 0.723±0.078 | 0.770±0.070 | 0.939±0.016 | Not Applicable | 49304 |
| DE-NRDE$_3$ | 4 | 0.421±0.074 | 0.206±0.096 | 0.281±0.091 | 0.583±0.086 | Not Applicable | 179371 |
| | 32 | 0.626±0.047 | 0.528±0.038 | 0.585±0.054 | 0.878±0.036 | Not Applicable | 179371 |
| | 128 | 0.841±0.042 | 0.810±0.029 | 0.840±0.038 | 0.960±0.022 | Not Applicable | 241223 |
| LORD$_{1\to2}$ | 4 | 0.795±0.051 | 0.744±0.051 | 0.787±0.053 | 0.948±0.009 | 24299 | 38973 |
| | 32 | 0.841±0.053 | 0.841±0.058 | 0.843±0.051 | 0.957±0.016 | 24299 | 38973 |
| | 128 | 0.759±0.064 | 0.748±0.082 | 0.760±0.061 | 0.900±0.009 | 24299 | 34813 |
| LORD$_{1\to3}$ | 4 | 0.774±0.088 | 0.733±0.122 | 0.759±0.101 | 0.944±0.023 | 86907 | 34813 |
| | 32 | 0.862±0.053 | 0.855±0.058 | 0.861±0.054 | 0.963±0.021 | 86907 | 34813 |
| | 128 | 0.744±0.081 | 0.716±0.087 | 0.737±0.076 | 0.892±0.044 | 86907 | 38973 |
| LORD$_{2\to3}$ | 4 | 0.821±0.054 | 0.783±0.062 | 0.815±0.056 | 0.960±0.011 | 278679 | 132465 |
| | 32 | 0.841±0.021 | 0.793±0.053 | 0.835±0.027 | 0.963±0.008 | 278679 | 132465 |
| | 128 | 0.841±0.038 | 0.823±0.055 | 0.835±0.049 | 0.949±0.023 | 278679 | 128305 |

Table 11: `CounterMovementJump`

| Method | $P$ | Accuracy | Macro F1 | Weighted F1 | ROCAUC | #Params(D) | #Params(R) |
|---|---|---|---|---|---|---|---|
| ODE-RNN | 64 | 0.474±0.023 | 0.453±0.027 | 0.455±0.026 | 0.589±0.018 | Not Applicable | 24737 |
| | 128 | 0.470±0.093 | 0.458±0.094 | 0.459±0.094 | 0.617±0.055 | Not Applicable | 213537 |
| | 512 | 0.449±0.032 | 0.436±0.037 | 0.436±0.039 | 0.593±0.031 | Not Applicable | 139425 |
| NCDE | 64 | 0.351±0.057 | 0.317±0.091 | 0.318±0.091 | 0.473±0.024 | Not Applicable | 58371 |
| | 128 | 0.404±0.064 | 0.350±0.050 | 0.352±0.050 | 0.520±0.007 | Not Applicable | 25475 |
| | 512 | 0.411±0.047 | 0.379±0.064 | 0.378±0.065 | 0.496±0.026 | Not Applicable | 46723 |
| ANCDE | 64 | 0.355±0.041 | 0.342±0.034 | 0.342±0.035 | 0.526±0.033 | Not Applicable | 135852 |
| | 128 | 0.467±0.060 | 0.457±0.065 | 0.457±0.065 | 0.604±0.031 | Not Applicable | 252332 |
| | 512 | 0.458±0.073 | 0.428±0.102 | 0.428±0.102 | 0.588±0.074 | Not Applicable | 285740 |
| NRDE$_2$ | 64 | 0.396±0.116 | 0.391±0.118 | 0.391±0.118 | 0.560±0.089 | Not Applicable | 107907 |
| | 128 | 0.396±0.076 | 0.379±0.079 | 0.379±0.078 | 0.513±0.052 | Not Applicable | 189059 |
| | 512 | 0.396±0.091 | 0.388±0.092 | 0.388±0.092 | 0.546±0.076 | Not Applicable | 50435 |
| NRDE$_3$ | 64 | 0.465±0.088 | 0.462±0.093 | 0.463±0.093 | 0.616±0.062 | Not Applicable | 529411 |
| | 128 | 0.409±0.084 | 0.399±0.081 | 0.400±0.081 | 0.573±0.070 | Not Applicable | 521859 |
| | 512 | 0.465±0.055 | 0.453±0.050 | 0.453±0.051 | 0.618±0.045 | Not Applicable | 2107395 |
| DE-NRDE$_2$ | 64 | 0.398±0.075 | 0.390±0.076 | 0.389±0.075 | 0.573±0.076 | Not Applicable | 51910 |
| | 128 | 0.384±0.058 | 0.334±0.071 | 0.334±0.071 | 0.567±0.054 | Not Applicable | 90886 |
| | 512 | 0.375±0.054 | 0.359±0.064 | 0.359±0.064 | 0.525±0.055 | Not Applicable | 39114 |
| DE-NRDE$_3$ | 64 | 0.409±0.072 | 0.373±0.073 | 0.373±0.074 | 0.568±0.068 | Not Applicable | 204300 |
| | 128 | 0.474±0.038 | 0.437±0.053 | 0.437±0.054 | 0.626±0.030 | Not Applicable | 279378 |
| | 512 | 0.425±0.069 | 0.396±0.076 | 0.398±0.076 | 0.571±0.059 | Not Applicable | 730892 |
| LORD$_{1\to2}$ | 64 | 0.589±0.038 | 0.580±0.041 | 0.580±0.041 | 0.717±0.023 | 3262 | 87191 |
| | 128 | 0.566±0.061 | 0.563±0.066 | 0.563±0.066 | 0.676±0.058 | 12158 | 51063 |
| | 512 | 0.501±0.080 | 0.495±0.082 | 0.495±0.082 | 0.638±0.053 | 7998 | 65399 |
| LORD$_{1\to3}$ | 64 | 0.555±0.041 | 0.551±0.042 | 0.550±0.042 | 0.688±0.038 | 7282 | 43127 |
| | 128 | 0.528±0.029 | 0.523±0.027 | 0.523±0.027 | 0.656±0.024 | 53746 | 137879 |
| | 512 | 0.508±0.033 | 0.499±0.028 | 0.499±0.028 | 0.637±0.020 | 8338 | 88439 |
| LORD$_{2\to3}$ | 64 | 0.396±0.049 | 0.383±0.052 | 0.383±0.052 | 0.540±0.061 | 77158 | 113265 |
| | 128 | 0.396±0.078 | 0.384±0.083 | 0.385±0.083 | 0.542±0.068 | 27110 | 146737 |
| | 512 | 0.400±0.044 | 0.374±0.022 | 0.375±0.022 | 0.537±0.047 | 31270 | 36369 |

# D FULL EXPERIMENTAL RESULTS

In Tables 10 to 15, all experimental results are shown. #Params(D) and #Params(R) denote the number of parameters in the decoder part and in the rest parts except the decoder, respectively.

Table 12: `SelfRegulationSCP2`

| Method | $P$ | Accuracy | F1 | ROCAUC | #Params(D) | #Params(R) |
|---|---|---|---|---|---|---|
| ODE-RNN | 32 | 0.551±0.054 | 0.540±0.064 | 0.603±0.041 | Not Applicable | 57375 |
| | 64 | 0.579±0.012 | 0.462±0.051 | 0.564±0.019 | Not Applicable | 40991 |
| | 256 | 0.558±0.045 | 0.480±0.039 | 0.533±0.055 | Not Applicable | 139295 |
| NCDE | 32 | 0.586±0.071 | 0.471±0.124 | 0.577±0.130 | Not Applicable | 42241 |
| | 64 | 0.516±0.079 | 0.459±0.156 | 0.557±0.129 | Not Applicable | 214657 |
| | 256 | 0.544±0.058 | 0.537±0.087 | 0.578±0.086 | Not Applicable | 316161 |
| ANCDE | 32 | 0.604±0.064 | 0.552±0.048 | 0.646±0.041 | Not Applicable | 391698 |
| | 64 | 0.544±0.039 | 0.490±0.048 | 0.575±0.074 | Not Applicable | 134034 |
| | 256 | 0.488±0.052 | 0.427±0.067 | 0.461±0.073 | Not Applicable | 208914 |
| $NRDE_2$ | 32 | 0.519±0.110 | 0.513±0.058 | 0.572±0.092 | Not Applicable | 322689 |
| | 64 | 0.561±0.090 | 0.449±0.062 | 0.575±0.071 | Not Applicable | 622209 |
| | 256 | 0.516±0.110 | 0.498±0.071 | 0.564±0.096 | Not Applicable | 622209 |
| $NRDE_3$ | 32 | 0.537±0.052 | 0.560±0.050 | 0.561±0.049 | Not Applicable | 3402753 |
| | 64 | 0.558±0.059 | 0.504±0.118 | 0.586±0.107 | Not Applicable | 1710977 |
| | 256 | 0.498±0.066 | 0.483±0.074 | 0.543±0.079 | Not Applicable | 3438465 |
| $DE-NRDE_2$ | 32 | 0.589±0.095 | 0.525±0.130 | 0.596±0.072 | Not Applicable | 111051 |
| | 64 | 0.551±0.063 | 0.496±0.081 | 0.566±0.073 | Not Applicable | 196747 |
| | 256 | 0.530±0.078 | 0.501±0.078 | 0.538±0.088 | Not Applicable | 196747 |
| $DE-NRDE_3$ | 32 | 0.551±0.109 | 0.570±0.074 | 0.655±0.076 | Not Applicable | 1092158 |
| | 64 | 0.505±0.076 | 0.539±0.059 | 0.566±0.110 | Not Applicable | 542462 |
| | 256 | 0.512±0.100 | 0.547±0.031 | 0.591±0.069 | Not Applicable | 1137022 |
| $LORD_{1\rightarrow2}$ | 32 | 0.570±0.105 | 0.566±0.058 | 0.652±0.059 | 14572 | 24361 |
| | 64 | 0.582±0.119 | 0.613±0.057 | 0.681±0.071 | 76684 | 139433 |
| | 256 | 0.530±0.081 | 0.512±0.081 | 0.536±0.087 | 76684 | 74857 |
| $LORD_{1\rightarrow3}$ | 32 | 0.572±0.058 | 0.501±0.056 | 0.584±0.036 | 70132 | 138761 |
| | 64 | 0.618±0.099 | 0.591±0.109 | 0.659±0.094 | 71188 | 123241 |
| | 256 | 0.526±0.105 | 0.558±0.038 | 0.607±0.061 | 278452 | 161161 |
| $LORD_{2\rightarrow3}$ | 32 | 0.607±0.059 | 0.552±0.087 | 0.645±0.093 | 260580 | 216405 |
| | 64 | 0.607±0.095 | 0.555±0.072 | 0.619±0.060 | 999684 | 534037 |
| | 256 | 0.596±0.075 | 0.529±0.055 | 0.623±0.064 | 259524 | 254549 |

Table 13: `BIDMC32HR`

| Method | $P$ | $R^2$ | Explained Variance | MSE | MAE | #Params(D) | #Params(R) |
|---|---|---|---|---|---|---|---|
| ODE-RNN | 8 | 0.569±0.290 | 0.570±0.290 | 0.415±0.279 | 0.457±0.172 | Not Applicable | 3871 |
| | 128 | 0.924±0.011 | 0.925±0.011 | 0.073±0.011 | 0.167±0.009 | Not Applicable | 15391 |
| | 512 | 0.811±0.024 | 0.811±0.025 | 0.182±0.024 | 0.285±0.019 | Not Applicable | 52255 |
| NCDE | 8 | 0.388±0.042 | 0.388±0.042 | 0.590±0.041 | 0.552±0.023 | Not Applicable | 49921 |
| | 128 | 0.227±0.042 | 0.231±0.046 | 0.745±0.040 | 0.638±0.017 | Not Applicable | 49921 |
| | 512 | 0.191±0.041 | 0.192±0.041 | 0.779±0.039 | 0.654±0.018 | Not Applicable | 49921 |
| ANCDE | 8 | 0.437±0.031 | 0.440±0.031 | 0.542±0.030 | 0.527±0.019 | Not Applicable | 47194 |
| | 128 | 0.173±0.062 | 0.174±0.062 | 0.796±0.060 | 0.666±0.019 | Not Applicable | 47194 |
| | 512 | 0.264±0.011 | 0.270±0.012 | 0.709±0.011 | 0.628±0.016 | Not Applicable | 47194 |
| $NRDE_2$ | 8 | 0.630±0.044 | 0.631±0.044 | 0.356±0.042 | 0.452±0.030 | Not Applicable | 74689 |
| | 128 | 0.665±0.072 | 0.665±0.072 | 0.323±0.069 | 0.409±0.047 | Not Applicable | 74689 |
| | 512 | 0.642±0.071 | 0.643±0.071 | 0.344±0.069 | 0.419±0.054 | Not Applicable | 74689 |
| $NRDE_3$ | 8 | 0.650±0.082 | 0.650±0.082 | 0.337±0.079 | 0.430±0.054 | Not Applicable | 140737 |
| | 128 | 0.582±0.131 | 0.583±0.131 | 0.402±0.126 | 0.458±0.085 | Not Applicable | 140737 |
| | 512 | 0.395±0.131 | 0.396±0.131 | 0.582±0.126 | 0.535±0.087 | Not Applicable | 140737 |
| $DE-NRDE_2$ | 8 | 0.822±0.077 | 0.822±0.077 | 0.171±0.074 | 0.279±0.080 | Not Applicable | 63045 |
| | 128 | 0.758±0.051 | 0.759±0.052 | 0.233±0.049 | 0.332±0.045 | Not Applicable | 59589 |
| | 512 | 0.640±0.024 | 0.643±0.023 | 0.346±0.023 | 0.412±0.016 | Not Applicable | 58885 |
| $DE-NRDE_3$ | 8 | 0.885±0.049 | 0.885±0.049 | 0.111±0.048 | 0.222±0.050 | Not Applicable | 119050 |
| | 128 | 0.927±0.016 | 0.928±0.016 | 0.070±0.016 | 0.176±0.024 | Not Applicable | 102538 |
| | 512 | 0.810±0.085 | 0.810±0.086 | 0.183±0.082 | 0.299±0.080 | Not Applicable | 102538 |
| $LORD_{1\rightarrow2}$ | 8 | 0.979±0.006 | 0.979±0.006 | 0.020±0.005 | 0.084±0.014 | 10285 | 27277 |
| | 128 | 0.937±0.009 | 0.937±0.009 | 0.061±0.008 | 0.119±0.005 | 3277 | 34189 |
| | 512 | 0.441±0.039 | 0.443±0.041 | 0.538±0.038 | 0.479±0.033 | 3277 | 34189 |
| $LORD_{1\rightarrow3}$ | 8 | 0.982±0.005 | 0.982±0.005 | 0.018±0.005 | 0.079±0.008 | 12645 | 27277 |
| | 128 | 0.922±0.012 | 0.923±0.013 | 0.075±0.012 | 0.129±0.013 | 40997 | 65293 |
| | 512 | 0.450±0.031 | 0.452±0.029 | 0.530±0.030 | 0.474±0.007 | 4613 | 54925 |
| $LORD_{2\rightarrow3}$ | 8 | 0.978±0.006 | 0.979±0.006 | 0.021±0.006 | 0.078±0.009 | 15471 | 35563 |
| | 128 | 0.954±0.006 | 0.954±0.006 | 0.045±0.006 | 0.121±0.007 | 6095 | 51915 |
| | 512 | 0.848±0.043 | 0.849±0.043 | 0.146±0.041 | 0.216±0.028 | 15471 | 54283 |

Table 14: `BIDMC32RR`

| Method | $P$ | $R^2$ | Explained Variance | MSE | MAE | #Params(D) | #Params(R) |
|---|---|---|---|---|---|---|---|
| ODE-RNN | 8 | 0.536±0.195 | 0.540±0.193 | 0.447±0.188 | 0.494±0.124 | Not Applicable | 53791 |
| | 128 | 0.721±0.010 | 0.722±0.011 | 0.269±0.010 | 0.360±0.006 | Not Applicable | 122911 |
| | 512 | 0.651±0.018 | 0.652±0.017 | 0.337±0.018 | 0.413±0.016 | Not Applicable | 344095 |
| NCDE | 8 | 0.215±0.028 | 0.215±0.028 | 0.757±0.027 | 0.663±0.010 | Not Applicable | 86913 |
| | 128 | 0.337±0.053 | 0.339±0.053 | 0.639±0.051 | 0.595±0.015 | Not Applicable | 86913 |
| | 512 | 0.236±0.026 | 0.237±0.026 | 0.736±0.025 | 0.626±0.008 | Not Applicable | 86913 |
| ANCDE | 8 | 0.298±0.062 | 0.302±0.064 | 0.677±0.060 | 0.625±0.033 | Not Applicable | 47194 |
| | 128 | 0.386±0.097 | 0.387±0.098 | 0.592±0.094 | 0.576±0.036 | Not Applicable | 47194 |
| | 512 | 0.270±0.028 | 0.271±0.028 | 0.704±0.027 | 0.619±0.010 | Not Applicable | 55514 |
| NRDE$_2$ | 8 | 0.274±0.037 | 0.274±0.037 | 0.700±0.035 | 0.637±0.019 | Not Applicable | 123969 |
| | 128 | 0.521±0.082 | 0.521±0.082 | 0.462±0.079 | 0.500±0.035 | Not Applicable | 123969 |
| | 512 | 0.498±0.162 | 0.498±0.162 | 0.484±0.156 | 0.512±0.075 | Not Applicable | 123969 |
| NRDE$_3$ | 8 | 0.337±0.035 | 0.338±0.035 | 0.639±0.034 | 0.600±0.019 | Not Applicable | 222785 |
| | 128 | 0.646±0.138 | 0.647±0.138 | 0.341±0.133 | 0.403±0.087 | Not Applicable | 222785 |
| | 512 | 0.424±0.130 | 0.424±0.130 | 0.556±0.126 | 0.537±0.071 | Not Applicable | 222785 |
| DE-NRDE$_2$ | 8 | 0.644±0.013 | 0.647±0.012 | 0.343±0.013 | 0.388±0.017 | Not Applicable | 88196 |
| | 128 | 0.584±0.061 | 0.586±0.060 | 0.401±0.059 | 0.440±0.036 | Not Applicable | 100677 |
| | 512 | 0.552±0.063 | 0.554±0.061 | 0.432±0.061 | 0.474±0.033 | Not Applicable | 99973 |
| DE-NRDE$_3$ | 8 | 0.699±0.055 | 0.699±0.055 | 0.290±0.053 | 0.351±0.041 | Not Applicable | 139144 |
| | 128 | 0.703±0.017 | 0.703±0.017 | 0.287±0.016 | 0.352±0.015 | Not Applicable | 164106 |
| | 512 | 0.574±0.053 | 0.575±0.054 | 0.411±0.051 | 0.454±0.025 | Not Applicable | 162570 |
| LORD$_{1\rightarrow2}$ | 8 | 0.808±0.021 | 0.809±0.022 | 0.185±0.020 | 0.272±0.011 | 10285 | 27277 |
| | 128 | 0.662±0.007 | 0.664±0.007 | 0.326±0.006 | 0.386±0.012 | 10285 | 27277 |
| | 512 | 0.448±0.044 | 0.450±0.044 | 0.532±0.042 | 0.522±0.012 | 10285 | 39757 |
| LORD$_{1\rightarrow3}$ | 8 | 0.798±0.017 | 0.799±0.017 | 0.195±0.016 | 0.277±0.014 | 12645 | 27277 |
| | 128 | 0.671±0.016 | 0.672±0.015 | 0.318±0.015 | 0.376±0.012 | 12645 | 44973 |
| | 512 | 0.451±0.026 | 0.453±0.023 | 0.530±0.025 | 0.520±0.012 | 12645 | 44973 |
| LORD$_{2\rightarrow3}$ | 8 | 0.800±0.007 | 0.801±0.007 | 0.192±0.007 | 0.281±0.008 | 15471 | 35563 |
| | 128 | 0.744±0.011 | 0.744±0.011 | 0.247±0.010 | 0.315±0.010 | 15471 | 136459 |
| | 512 | 0.662±0.015 | 0.664±0.014 | 0.326±0.014 | 0.386±0.011 | 46511 | 103531 |

Table 15: `BIDMC32SpO2`

| Method | $P$ | $R^2$ | Explained Variance | MSE | MAE | #Params(D) | #Params(R) |
|---|---|---|---|---|---|---|---|
| ODE-RNN | 8 | 0.551±0.108 | 0.552±0.108 | 0.464±0.112 | 0.487±0.082 | Not Applicable | 3871 |
| | 128 | 0.899±0.025 | 0.901±0.023 | 0.104±0.026 | 0.225±0.035 | Not Applicable | 15391 |
| | 512 | 0.749±0.024 | 0.749±0.024 | 0.259±0.025 | 0.365±0.021 | Not Applicable | 52255 |
| NCDE | 8 | 0.240±0.071 | 0.245±0.067 | 0.785±0.073 | 0.686±0.034 | Not Applicable | 86913 |
| | 128 | 0.215±0.074 | 0.217±0.072 | 0.811±0.077 | 0.694±0.035 | Not Applicable | 86913 |
| | 512 | 0.328±0.046 | 0.332±0.045 | 0.694±0.046 | 0.639±0.007 | Not Applicable | 86913 |
| ANCDE | 8 | 0.311±0.034 | 0.313±0.035 | 0.712±0.035 | 0.662±0.014 | Not Applicable | 47194 |
| | 128 | 0.327±0.049 | 0.327±0.049 | 0.696±0.050 | 0.631±0.017 | Not Applicable | 47194 |
| | 512 | 0.372±0.033 | 0.374±0.034 | 0.648±0.034 | 0.592±0.013 | Not Applicable | 47194 |
| NRDE$_2$ | 8 | 0.218±0.078 | 0.219±0.078 | 0.807±0.080 | 0.694±0.036 | Not Applicable | 123969 |
| | 128 | 0.470±0.100 | 0.471±0.100 | 0.547±0.103 | 0.559±0.048 | Not Applicable | 123969 |
| | 512 | 0.607±0.180 | 0.607±0.180 | 0.406±0.186 | 0.455±0.123 | Not Applicable | 123969 |
| NRDE$_3$ | 8 | 0.131±0.091 | 0.134±0.090 | 0.898±0.094 | 0.720±0.027 | Not Applicable | 222785 |
| | 128 | 0.681±0.147 | 0.681±0.147 | 0.330±0.151 | 0.410±0.126 | Not Applicable | 222785 |
| | 512 | 0.596±0.181 | 0.596±0.181 | 0.417±0.187 | 0.463±0.139 | Not Applicable | 222785 |
| DE-NRDE$_2$ | 8 | 0.854±0.013 | 0.854±0.013 | 0.151±0.013 | 0.252±0.015 | Not Applicable | 88196 |
| | 128 | 0.740±0.047 | 0.740±0.048 | 0.269±0.049 | 0.313±0.039 | Not Applicable | 99973 |
| | 512 | 0.586±0.042 | 0.586±0.042 | 0.428±0.043 | 0.453±0.033 | Not Applicable | 99973 |
| DE-NRDE$_3$ | 8 | 0.903±0.079 | 0.903±0.079 | 0.100±0.082 | 0.192±0.090 | Not Applicable | 139144 |
| | 128 | 0.926±0.026 | 0.927±0.026 | 0.076±0.027 | 0.164±0.024 | Not Applicable | 162570 |
| | 512 | 0.759±0.067 | 0.761±0.067 | 0.249±0.069 | 0.357±0.061 | Not Applicable | 162570 |
| LORD$_{1\rightarrow2}$ | 8 | 0.975±0.003 | 0.976±0.002 | 0.025±0.003 | 0.096±0.009 | 10285 | 27277 |
| | 128 | 0.909±0.009 | 0.910±0.009 | 0.094±0.009 | 0.172±0.013 | 3277 | 34189 |
| | 512 | 0.583±0.047 | 0.583±0.047 | 0.431±0.048 | 0.435±0.024 | 10285 | 61549 |
| LORD$_{1\rightarrow3}$ | 8 | 0.981±0.005 | 0.981±0.004 | 0.020±0.005 | 0.086±0.006 | 12645 | 27277 |
| | 128 | 0.889±0.025 | 0.889±0.025 | 0.115±0.026 | 0.186±0.029 | 12645 | 27277 |
| | 512 | 0.521±0.045 | 0.522±0.044 | 0.495±0.046 | 0.459±0.024 | 4613 | 38349 |
| LORD$_{2\rightarrow3}$ | 8 | 0.981±0.005 | 0.981±0.005 | 0.019±0.005 | 0.088±0.007 | 15471 | 35563 |
| | 128 | 0.940±0.007 | 0.940±0.007 | 0.062±0.007 | 0.145±0.004 | 15471 | 56395 |
| | 512 | 0.829±0.027 | 0.830±0.027 | 0.177±0.028 | 0.236±0.016 | 46511 | 69323 |

Table 16: Sensitivity of $max\_iter_{AE}$ in `EigenWorms` ($P = 128$)

| Method | $max\_iter_{AE}$ | Accuracy | Macro F1 | Weighted F1 | ROCAUC |
|---|---|---|---|---|---|
| $\text{LORD}_{1\rightarrow 2}$ | 0 | 0.656±0.094 | 0.588±0.107 | 0.649±0.100 | 0.847±0.056 |
| | 100 | 0.713±0.042 | 0.716±0.058 | 0.713±0.046 | 0.862±0.023 |
| | 300 | 0.744±0.048 | 0.727±0.059 | 0.744±0.051 | 0.893±0.029 |
| | 500 | 0.733±0.105 | 0.725±0.104 | 0.738±0.111 | 0.889±0.035 |
| | 700 | 0.774±0.049 | 0.773±0.070 | 0.777±0.056 | 0.908±0.026 |
| | 900 | 0.723±0.091 | 0.702±0.092 | 0.722±0.083 | 0.891±0.054 |
| $\text{LORD}_{1\rightarrow 3}$ | 0 | 0.600±0.140 | 0.513±0.215 | 0.561±0.188 | 0.820±0.068 |
| | 100 | 0.713±0.056 | 0.636±0.139 | 0.694±0.085 | 0.884±0.030 |
| | 300 | 0.733±0.069 | 0.725±0.053 | 0.730±0.063 | 0.868±0.036 |
| | 500 | 0.733±0.039 | 0.722±0.054 | 0.730±0.046 | 0.872±0.035 |
| | 700 | 0.744±0.031 | 0.730±0.043 | 0.741±0.034 | 0.872±0.021 |
| | 900 | 0.744±0.048 | 0.724±0.040 | 0.740±0.053 | 0.884±0.015 |
| $\text{LORD}_{2\rightarrow 3}$ | 0 | 0.477±0.059 | 0.338±0.105 | 0.420±0.078 | 0.764±0.051 |
| | 100 | 0.667±0.091 | 0.591±0.127 | 0.642±0.106 | 0.888±0.044 |
| | 300 | 0.790±0.042 | 0.754±0.060 | 0.783±0.050 | 0.927±0.017 |
| | 500 | 0.826±0.071 | 0.789±0.113 | 0.825±0.072 | 0.951±0.029 |
| | 700 | 0.846±0.101 | 0.816±0.126 | 0.847±0.103 | 0.928±0.046 |
| | 900 | 0.856±0.067 | 0.833±0.087 | 0.857±0.067 | 0.949±0.027 |

# E  SENSITIVITY EXPERIMENTS

Fig. 6 and Table 16 show that in general, the performance of `LORD` is improved as $max\_iter_{AE}$ gets larger in `EigenWorms` ($P = 128$). Our encoder-decoder structure can successfully integrate the lower-depth log-signature and the higher-depth log-signature information into the embedding vector **e**, and the well-integrated information helps the model perform better. In general, the lower-depth and the higher-depth log-signature information are blended better as the encoder-decoder structure is more trained.

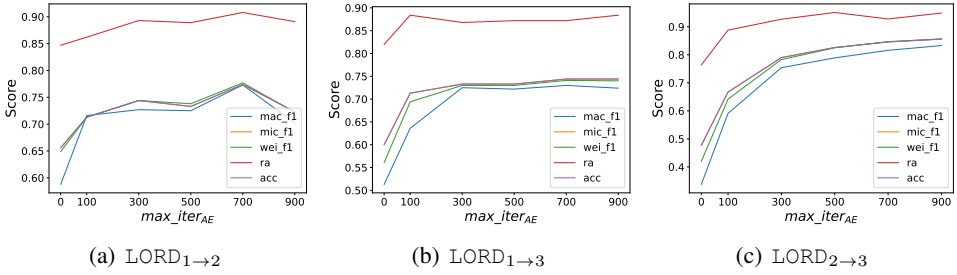

(a) $\text{LORD}_{1\rightarrow 2}$      (b) $\text{LORD}_{1\rightarrow 3}$      (c) $\text{LORD}_{2\rightarrow 3}$

Figure 6: Sensitivity to $max\_iter_{AE}$

# F  TRAIN LOSS

Fig. 7 visualizes several training cases for our method and the original NRDE design. In general, our method shows much better stability across the entire training period.

# G  END-TO-END TRAINING

`LORD` has two distinguished characteristics in its training process. First, `LORD` uses a two-phase training strategy, a pre-training and a main training processes. Second, the training process of `LORD` is not end-to-end. In other words, the decoder is not used and the encoder is fixed in the main training phase. In this section, we test with various end-to-end-training settings for `LORD`.

There are three possible configurations for the end-to-end-training. The first configuration is training the encoder (rather than fixing it) with $L_{TASK}$ in the main training phase, which is denoted as `FineTuning`. The second one is `Co-Train` where both $L_{AE}$ and $L_{TASK}$ are used for training the encoder, the decoder and the main NRDE — recall that in `FineTuning`, we use only $L_{TASK}$

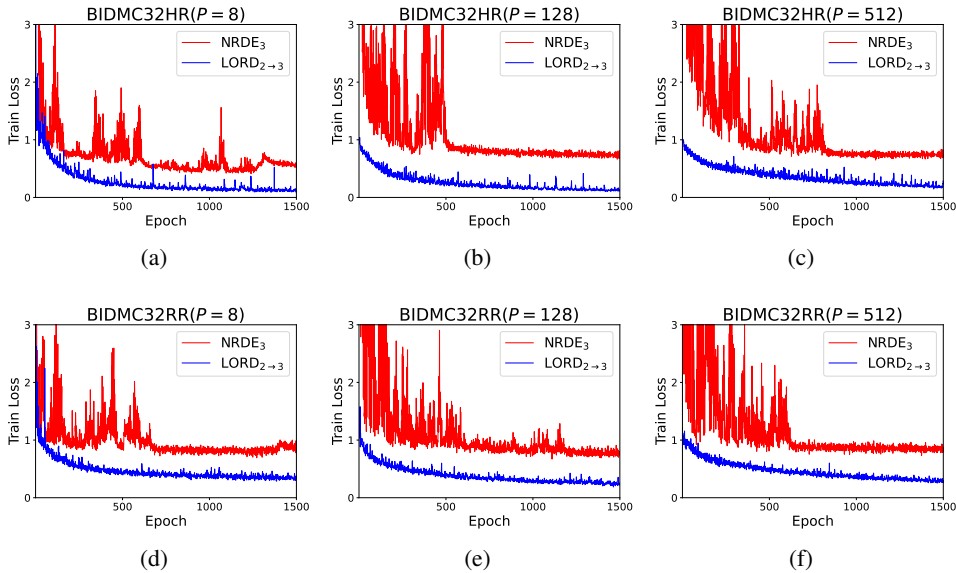

Figure 7: Train loss of $\mathtt{NRDE}_3$ and $\mathtt{LORD}_{2\to3}$

Table 17: Comparison between $\mathtt{LORD}$ and its various end-to-end training variations in $\mathtt{EigenWorms}$ ($P = 128$)

|  | Method | Accuracy | Macro F1 | Weighted F1 | ROCAUC |
|---|---|---|---|---|---|
| $\mathtt{LORD}_{1\to2}$ | LORD | 0.759±0.064 | 0.748±0.082 | 0.760±0.061 | 0.900±0.009 |
|  | FineTuning | 0.744±0.075 | 0.726±0.086 | 0.740±0.077 | 0.882±0.042 |
|  | Co-Train | 0.718±0.021 | 0.686±0.025 | 0.715±0.024 | 0.865±0.030 |
|  | Co-Train(w.o. pre) | 0.679±0.033 | 0.650±0.031 | 0.669±0.029 | 0.832±0.013 |
| $\mathtt{LORD}_{1\to3}$ | LORD | 0.744±0.081 | 0.716±0.087 | 0.737±0.076 | 0.892±0.044 |
|  | FineTuning | 0.692±0.055 | 0.649±0.049 | 0.680±0.069 | 0.865±0.018 |
|  | Co-Train | 0.744±0.036 | 0.723±0.059 | 0.741±0.037 | 0.868±0.022 |
|  | Co-Train(w.o. pre) | 0.692±0.069 | 0.674±0.083 | 0.694±0.064 | 0.839±0.066 |
| $\mathtt{LORD}_{2\to3}$ | LORD | **0.841±0.038** | **0.823±0.055** | **0.835±0.049** | **0.949±0.023** |
|  | FineTuning | 0.731±0.061 | 0.687±0.075 | 0.714±0.062 | 0.919±0.005 |
|  | Co-Train | 0.808±0.061 | 0.772±0.074 | 0.803±0.071 | 0.948±0.021 |
|  | Co-Train(w.o. pre) | 0.583±0.112 | 0.499±0.145 | 0.557±0.117 | 0.804±0.065 |

to train the encoder. The last method is `Co-Train(w.o. pre)` which is the same as `Co-Train` without any pre-training process. The same hyperparameters are used for all end-to-end models.

We experiment with `EigenWorms` ($P = 128$). Table 16 shows the results. Among the three end-to-end models, `Co-Train` shows the best performance. `Co-Train(w.o. pre)` has the worst performance in general. These results prove that the pre-training makes the main training easier.

Table 18: The best hyperparameter in `ODE-RNN`

| Data | $P$ | Hidden Path Size |
|---|---|---|
| Character-Trajectory | 1 | 128 |
|  | 4 | 128 |
|  | 16 | 256 |
|  | 32 | 64 |
| LiveFuel-MoistureContent | 1 | 128 |
|  | 4 | 64 |
|  | 16 | 64 |
|  | 32 | 128 |

Table 19: The best hyperparameter in `NCDE` and `NRDE`

| Data | $P$ | Hidden Path Size $D=1$ | 2 | 3 | CDE Function's Hidden Size 1 | 2 | 3 | #Hidden Layers 1 | 2 | 3 |
|---|---|---|---|---|---|---|---|---|---|---|
| Character-Trajectory | 1 | 64 | - | - | 256 | - | - | 3 | - | - |
|  | 4 | 128 | 256 | 64 | 256 | 256 | 128 | 3 | 3 | 2 |
|  | 16 | 256 | 256 | 128 | 64 | 64 | 256 | 2 | 3 | 2 |
|  | 32 | 256 | 128 | 128 | 64 | 64 | 128 | 2 | 3 | 2 |
| LiveFuel-MoistureContent | 1 | 64 | - | - | 64 | - | - | 3 | - | - |
|  | 4 | 64 | 128 | 128 | 128 | 256 | 64 | 3 | 3 | 2 |
|  | 16 | 64 | 256 | 64 | 64 | 256 | 256 | 2 | 3 | 2 |
|  | 32 | 256 | 256 | 64 | 256 | 128 | 64 | 3 | 3 | 2 |

Table 20: The best hyperparameter in `ANCDE`

| Data | $P$ | Hidden Path Size | #Hidden Layers | Attention Channel Size |
|---|---|---|---|---|
| Character-Trajectory | 1 | 256 | 3 | 256 |
| | 4 | 256 | 2 | 64 |
| | 16 | 256 | 3 | 64 |
| | 32 | 256 | 2 | 64 |
| LiveFuel-MoistureContent | 1 | 128 | 3 | 64 |
| | 4 | 128 | 3 | 128 |
| | 16 | 128 | 2 | 64 |
| | 32 | 256 | 3 | 64 |

Table 21: The best hyperparameter in `DE-NRDE`

| Data | $P$ | Compression Ratio | | #Hidden Layers | | Hidden Size | |
|---|---|---|---|---|---|---|---|
| | | $D=2$ | 3 | 2 | 3 | 2 | 3 |
| Character-Trajectory | 4 | 0.7 | 0.3 | 1 | 1 | 64 | 128 |
| | 16 | 0.5 | 0.5 | 1 | 1 | 128 | 64 |
| | 32 | 0.7 | 0.5 | 1 | 2 | 128 | 128 |
| LiveFuel-MoistureContent | 4 | 0.7 | 0.7 | 2 | 2 | 64 | 128 |
| | 16 | 0.3 | 0.3 | 2 | 1 | 64 | 64 |
| | 32 | 0.7 | 0.5 | 2 | 2 | 128 | 64 |

Table 22: The best hyperparameter in `LORD`

| Method | Data | $P$ | $N_f$ | $N_g$ | $N_o$ | $h_f$ | $h_g$ | $h_o$ | $c_{AE}$ | $c_{TASK}$ | $c_e$ | $max\_iter$ AE | TASK |
|---|---|---|---|---|---|---|---|---|---|---|---|---|---|
| LORD$_{1\rightarrow2}$ | Character-Trajectory | 4 | 3 | 3 | 3 | 64 | 256 | 64 | $1\times10^{-6}$ | $1\times10^{-6}$ | 0 | 1000 | 400 |
| | | 16 | 3 | 3 | 3 | 64 | 128 | 64 | $1\times10^{-6}$ | $1\times10^{-6}$ | 0 | 1000 | 400 |
| | | 32 | 3 | 3 | 3 | 128 | 256 | 128 | $1\times10^{-6}$ | $1\times10^{-6}$ | 0 | 1000 | 400 |
| | LiveFuel-MoistureContent | 4 | 3 | 3 | 3 | 128 | 128 | 128 | $1\times10^{-6}$ | $1\times10^{-6}$ | 0 | 2000 | 400 |
| | | 16 | 3 | 3 | 3 | 256 | 256 | 256 | $1\times10^{-6}$ | $1\times10^{-6}$ | 0 | 2000 | 400 |
| | | 32 | 3 | 3 | 3 | 256 | 256 | 256 | $1\times10^{-6}$ | $1\times10^{-6}$ | 0 | 1500 | 400 |
| LORD$_{1\rightarrow3}$ | Character-Trajectory | 4 | 3 | 3 | 3 | 128 | 128 | 128 | $1\times10^{-6}$ | $1\times10^{-6}$ | 0 | 1000 | 400 |
| | | 16 | 3 | 3 | 3 | 128 | 128 | 128 | $1\times10^{-6}$ | $1\times10^{-6}$ | 0 | 1000 | 400 |
| | | 32 | 3 | 3 | 3 | 128 | 64 | 128 | $1\times10^{-6}$ | $1\times10^{-6}$ | 0 | 1000 | 400 |
| | LiveFuel-MoistureContent | 4 | 3 | 3 | 3 | 256 | 256 | 256 | $1\times10^{-6}$ | $1\times10^{-6}$ | 0 | 2000 | 400 |
| | | 16 | 3 | 3 | 3 | 64 | 64 | 64 | $1\times10^{-6}$ | $1\times10^{-6}$ | 0 | 2000 | 400 |
| | | 32 | 3 | 3 | 3 | 128 | 64 | 128 | $1\times10^{-6}$ | $1\times10^{-6}$ | 0 | 1500 | 400 |
| LORD$_{2\rightarrow3}$ | Character-Trajectory | 4 | 3 | 3 | 3 | 128 | 256 | 128 | $1\times10^{-6}$ | $1\times10^{-6}$ | 0 | 1000 | 400 |
| | | 16 | 3 | 3 | 3 | 256 | 256 | 256 | $1\times10^{-6}$ | $1\times10^{-6}$ | 0 | 1000 | 400 |
| | | 32 | 3 | 3 | 3 | 128 | 256 | 128 | $1\times10^{-6}$ | $1\times10^{-6}$ | 0 | 1000 | 400 |
| | LiveFuel-MoistureContent | 4 | 3 | 3 | 3 | 256 | 64 | 256 | $1\times10^{-6}$ | $1\times10^{-6}$ | 0 | 2000 | 400 |
| | | 16 | 3 | 3 | 3 | 128 | 256 | 128 | $1\times10^{-6}$ | $1\times10^{-6}$ | 0 | 2000 | 400 |
| | | 32 | 3 | 3 | 3 | 64 | 256 | 64 | $1\times10^{-6}$ | $1\times10^{-6}$ | 0 | 1500 | 400 |

## H ADDITIONAL EXPERIMENTS WITH SHORT TIME-SERIES

In this section, we experiment with short time-series datasets. We use `CharacterTrajectory` and `LiveFuelMoistureContent` from (Tan & Webb). The object of `Character-Trajectory` is to classify 20 characters using the trajectory information of writing on a tablet. Its length is 182. `LiveFuelMoistureContent` consists of daily reflectance data at 7 spectral bands for predicting moisture rate in vegetation. Its length is 365. In Tables 18 to 22, we summarize hyperparameters. Because of the short time-series lengths, we test $P=1$ for all except `NRDE`-based models.

In Table 23, `NRDE` achieves the best scores among baseline models and `LORD` achieves the best scores among all methods. In Table 24, `DE-NRDE`$_2$ is the best. Except `DE-NRDE`$_2$, `ODE-RNN` is the best. However, `LORD` also has comparable scores. From these results, `NRDE` and `NRDE`-based models are empirically useful for short-time series as well.

Table 23: `CharacterTrajectory`

| Method | $P$ | Accuracy | Macro F1 | Weighted F1 | ROCAUC | #Params(D) | #Params(R) |
|---|---|---|---|---|---|---|---|
| ODE-RNN | 1 | 0.716±0.036 | 0.697±0.041 | 0.712±0.038 | 0.969±0.007 | Not Applicable | 27570 |
| | 4 | 0.950±0.012 | 0.947±0.013 | 0.950±0.012 | 0.998±0.001 | Not Applicable | 29106 |
| | 16 | 0.968±0.004 | 0.967±0.005 | 0.968±0.004 | **0.999±0.000**$^*$ | Not Applicable | 103218 |
| | 32 | 0.962±0.009 | 0.959±0.010 | 0.962±0.009 | 0.997±0.001 | Not Applicable | 17650 |
| NCDE | 1 | 0.954±0.004 | 0.952±0.004 | 0.954±0.004 | 0.998±0.000 | Not Applicable | 149844 |
| | 4 | 0.951±0.005 | 0.948±0.005 | 0.951±0.005 | 0.998±0.000 | Not Applicable | 233620 |
| | 16 | 0.905±0.020 | 0.901±0.020 | 0.905±0.020 | 0.994±0.002 | Not Applicable | 93588 |
| | 32 | 0.829±0.012 | 0.818±0.013 | 0.828±0.012 | 0.983±0.002 | Not Applicable | 93588 |
| ANCDE | 1 | 0.959±0.007 | 0.958±0.007 | 0.959±0.007 | 0.998±0.000 | Not Applicable | 669757 |
| | 4 | 0.952±0.008 | 0.950±0.008 | 0.952±0.008 | 0.998±0.000 | Not Applicable | 286845 |
| | 16 | 0.897±0.009 | 0.893±0.011 | 0.896±0.010 | 0.994±0.001 | Not Applicable | 538173 |
| | 32 | 0.777±0.020 | 0.762±0.020 | 0.774±0.021 | 0.981±0.001 | Not Applicable | 103293 |
| NRDE$_2$ | 4 | 0.970±0.004 | 0.968±0.005 | 0.970±0.004 | **0.999±0.000**$^*$ | Not Applicable | 795924 |
| | 16 | 0.951±0.008 | 0.949±0.009 | 0.951±0.008 | 0.998±0.001 | Not Applicable | 193428 |
| | 32 | 0.950±0.007 | 0.948±0.008 | 0.950±0.007 | 0.998±0.000 | Not Applicable | 98836 |
| NRDE$_3$ | 4 | 0.966±0.006 | 0.965±0.006 | 0.966±0.006 | **0.999±0.000**$^*$ | Not Applicable | 274132 |
| | 16 | 0.960±0.006 | 0.960±0.006 | 0.961±0.006 | **0.999±0.000**$^*$ | Not Applicable | 531604 |
| | 32 | 0.961±0.002 | 0.959±0.002 | 0.961±0.002 | **0.999±0.000**$^*$ | Not Applicable | 1088916 |
| DE-NRDE$_2$ | 4 | 0.898±0.077 | 0.890±0.084 | 0.897±0.078 | 0.994±0.007 | Not Applicable | 599707 |
| | 16 | 0.907±0.013 | 0.900±0.012 | 0.906±0.012 | 0.996±0.002 | Not Applicable | 112281 |
| | 32 | 0.897±0.018 | 0.890±0.018 | 0.896±0.017 | 0.995±0.001 | Not Applicable | 76187 |
| DE-NRDE$_3$ | 4 | 0.933±0.031 | 0.926±0.035 | 0.932±0.032 | 0.997±0.001 | Not Applicable | 105885 |
| | 16 | 0.947±0.014 | 0.943±0.015 | 0.947±0.014 | 0.998±0.000 | Not Applicable | 286883 |
| | 32 | 0.913±0.013 | 0.907±0.014 | 0.912±0.013 | 0.996±0.001 | Not Applicable | 617891 |
| LORD$_{1\to2}$ | 4 | **0.977±0.002**$^*$ | **0.976±0.003**$^*$ | **0.977±0.002**$^*$ | **0.999±0.000**$^*$ | 12158 | 235880 |
| | 16 | 0.955±0.010 | 0.953±0.010 | 0.955±0.010 | 0.998±0.001 | 12158 | 96232 |
| | 32 | 0.829±0.069 | 0.817±0.077 | 0.826±0.074 | 0.985±0.011 | 40126 | 261928 |
| LORD$_{1\to3}$ | 4 | 0.976±0.005 | 0.975±0.006 | 0.976±0.005 | **0.999±0.000**$^*$ | 53746 | 98568 |
| | 16 | 0.959±0.006 | 0.957±0.006 | 0.959±0.006 | 0.998±0.001 | 53746 | 98568 |
| | 32 | 0.810±0.069 | 0.796±0.074 | 0.808±0.071 | 0.983±0.013 | 53746 | 63560 |
| LORD$_{2\to3}$ | 4 | 0.971±0.008 | 0.969±0.009 | 0.971±0.008 | **0.999±0.000**$^*$ | 77158 | 372418 |
| | 16 | **0.971±0.003** | **0.969±0.003** | **0.971±0.003** | **0.999±0.000**$^*$ | 218086 | 391650 |
| | 32 | **0.964±0.008** | **0.963±0.008** | **0.964±0.008** | **0.999±0.000**$^*$ | 77158 | 372418 |

Table 24: `LiveFuelMoistureContent`

| Method | $P$ | $R^2$ | Explained Variance | MSE | MAE | #Params(D) | #Params(R) |
|---|---|---|---|---|---|---|---|
| ODE-RNN | 1 | 0.026±0.005 | 0.026±0.005 | 1.064±0.005 | 0.775±0.001 | Not Applicable | 25631 |
| | 4 | 0.029±0.003 | 0.029±0.003 | 1.062±0.003 | 0.774±0.002 | Not Applicable | 10271 |
| | 16 | 0.032±0.003 | 0.033±0.003 | 1.058±0.004 | 0.775±0.004 | Not Applicable | 16415 |
| | 32 | 0.031±0.004 | 0.031±0.004 | 1.059±0.004 | 0.774±0.003 | Not Applicable | 57375 |
| NCDE | 1 | -0.009±0.037 | -0.008±0.037 | 1.102±0.040 | 0.787±0.010 | Not Applicable | 42241 |
| | 4 | -0.004±0.020 | -0.003±0.020 | 1.098±0.022 | 0.785±0.007 | Not Applicable | 91521 |
| | 16 | -0.043±0.031 | -0.043±0.031 | 1.140±0.034 | 0.794±0.011 | Not Applicable | 42241 |
| | 32 | 0.009±0.029 | 0.009±0.029 | 1.083±0.032 | 0.779±0.005 | Not Applicable | 660481 |
| ANCDE | 1 | -0.025±0.078 | -0.024±0.079 | 1.120±0.086 | 0.785±0.009 | Not Applicable | 101714 |
| | 4 | 0.014±0.007 | 0.015±0.007 | 1.077±0.007 | 0.781±0.003 | Not Applicable | 241938 |
| | 16 | 0.003±0.030 | 0.004±0.030 | 1.089±0.033 | 0.781±0.010 | Not Applicable | 93394 |
| | 32 | 0.006±0.017 | 0.006±0.017 | 1.087±0.019 | 0.784±0.009 | Not Applicable | 177746 |
| NRDE$_2$ | 4 | -0.010±0.043 | -0.009±0.043 | 1.103±0.047 | 0.785±0.004 | Not Applicable | 1284353 |
| | 16 | -0.373±0.274 | -0.373±0.274 | 1.501±0.299 | 0.824±0.020 | Not Applicable | 2502657 |
| | 32 | -0.151±0.166 | -0.151±0.166 | 1.258±0.182 | 0.816±0.023 | Not Applicable | 1240833 |
| NRDE$_3$ | 4 | -0.480±0.311 | -0.480±0.311 | 1.618±0.340 | 0.830±0.012 | Not Applicable | 1710977 |
| | 16 | -1.308±2.265 | -1.307±2.264 | 2.522±2.475 | 0.863±0.089 | Not Applicable | 3438465 |
| | 32 | -2.183±4.622 | -2.182±4.621 | 3.478±5.051 | 0.896±0.174 | Not Applicable | 857601 |
| DE-NRDE$_2$ | 4 | **0.040±0.007** | **0.040±0.007** | **1.049±0.007** | **0.770±0.005** | Not Applicable | 930650 |
| | 16 | **0.042±0.006** | **0.042±0.006** | **1.047±0.007** | **0.768±0.005** | Not Applicable | 799243 |
| | 32 | **0.043±0.005**$^*$ | **0.043±0.005**$^*$ | **1.046±0.006**$^*$ | **0.767±0.006**$^*$ | Not Applicable | 902042 |
| DE-NRDE$_3$ | 4 | -0.778±1.599 | -0.776±1.599 | 1.943±1.747 | 0.796±0.039 | Not Applicable | 1256207 |
| | 16 | -0.037±0.046 | -0.036±0.046 | 1.133±0.051 | 0.790±0.010 | Not Applicable | 1103486 |
| | 32 | -0.657±0.831 | -0.651±0.822 | 1.810±0.908 | 0.829±0.045 | Not Applicable | 457191 |
| LORD$_{1\to2}$ | 4 | 0.017±0.015 | 0.018±0.014 | 1.074±0.016 | 0.784±0.011 | 76684 | 120937 |
| | 16 | 0.019±0.007 | 0.020±0.008 | 1.072±0.008 | 0.781±0.007 | 216844 | 441481 |
| | 32 | 0.013±0.007 | 0.014±0.007 | 1.078±0.007 | 0.784±0.002 | 216844 | 441481 |
| LORD$_{1\to3}$ | 4 | 0.018±0.010 | 0.018±0.011 | 1.074±0.011 | 0.782±0.006 | 612148 | 441481 |
| | 16 | 0.002±0.009 | 0.003±0.009 | 1.091±0.010 | 0.790±0.003 | 136180 | 70153 |
| | 32 | 0.005±0.011 | 0.005±0.010 | 1.087±0.012 | 0.788±0.007 | 278452 | 99529 |
| LORD$_{2\to3}$ | 4 | -0.025±0.037 | -0.025±0.038 | 1.120±0.041 | 0.793±0.004 | 2081028 | 648629 |
| | 16 | -0.015±0.027 | -0.014±0.027 | 1.109±0.030 | 0.793±0.004 | 1016196 | 957557 |
| | 32 | 0.008±0.002 | 0.008±0.002 | 1.085±0.003 | 0.788±0.002 | 508356 | 540181 |

