# OpenReview forum: "LORD: Lower-Dimensional Embedding of Log-Signature in Neural Rough Differential Equations"
_ICLR.cc/2022/Conference — ICLR 2022 Poster_

### Official Review · Reviewer_yePm · 2021-10-17

**Correctness:** 4
**Technical Novelty And Significance:** 3
**Empirical Novelty And Significance:** 4
**Recommendation:** 8
**Confidence:** 5

**Main Review:**

## Discussion

**Main ideas** From an efficiency standpoint, I really like the idea of using a pretrained lightweight encoder / heavyweight decoder. Amortising costs like this is an eminently sensible thing to do.

I would note that I do not think this idea is entirely novel. For any supervised problem learning $x \mapsto y$, then it is very standard to perform dimensionality reduction (either linearly via PCA or nonlinearly via an autoencoder) on $x$ prior to training. The proposed technique is an instance of this idea.

Moreover I note that the encoder is evaluated on just the low-dimensional log-signatures, so that in principle the information contained in the higher-dimensional terms of the log-signatures has already been lost. Furthermore (in principle) the composed "main NRDE"-encoder could be learnt by a single NRDE as originally introduced.

Despite these (substantial) concerns, I am broadly in favour of this paper. The context in which these techniques is applied (long time series, rough differential equations, log-signatures) is already highly nontrivial, and the gains demonstrated are substantial. Moreover a great many papers are written by combining two existing techniques (here, NRDEs and autoencoders), and identifying profitable such combinations is very much worthwhile.

**Clarity** One concern I do have is the clarity of the presentation. I've collected some specific thoughts below, but I believe the quality of the writing can be improved in general, as the mathematical constructions used are generally not too clear. (I would suggest that the authors replicate this by sitting down with a colleague unfamiliar with this work, and have them highlight the points at which the constructions used are no longer clear.)

**Experiments** The experiments are very thorough, including all possible NCDE-derived benchmarks (at least that I am aware of), including ODE-RNNs, NCDEs, NRDEs, the attentive NCDEs of Jhin et al. 2021, and even a "naive" alternative to their own work. They consider six different datasets across both classification and regression, and both point-predictions and forecasting. They cover a wide variety of hyperparameters and several evaluation metrics.

I particularly appreciated the use of summary tables (Tables 2 and 3) as a readable summary of results. (I would suggest perhaps bolding the best value in each column?)

**Backpropagation** One important detail that I did not see discussed is the manner of backpropagation. Was this done via optimise-then-discretise (sometimes called just "the adjoint method") or via discretise-then-optimise ("through the solver")? (Or indeed via any other technique?) This can affect results quite dramatically. DtO is preferred if possible but Morrill et al. 2021 report requiring OtD due to memory constraints.

**The inclusion of time** I do want to highlight one mathematical typo, that is subtle but very important. On page 2, the interpolated path $X$ is claimed to satisfy $X(t_i) = x_i$, rather than (as is correct) $X(t_i) = (t_i, x_i)$.

- Suppose the contrary, and that only $X(t_i) = x_i$. Consider training an NCDE/NRDE just to output the timespan $t_n - t_1$ of a time series $((t_1, x_1), \ldots, (t_n, x_n))$. If all $x_i = 0$ then any reasonable interpolation scheme will produce $X \equiv 0$ and so  $z(t) = z(t_1) + \int_{t_1}^t f(z(s)) d X(s) = z(t_1)$ for all $t$. This is certainly not capable of calculating the timespan $t_n - t_1$.
- In contrast if time is present in the control ($X(t_i) = (t_i, x_i)$)  then we can take $f$ to be a matrix $[[1, 0], [0, 0]]$ so that $z(t) = z(t_1) + \int_{t_1}^t f(z(s))dX(s) = \int_{t_1}^t d s = z(t_1) + t - t_1$. (And then take the initial value network for $z(t_1)$ to be identically zero.)
- The universal approximation theorem for CDEs (Kidger et al. 2020, Appendix B) assures us that nothing else is missed, and that by including time in the control we are not losing any information unnecessarily.

## Specific points / typos

- Page 2: "calculate" -> "calculated"
- Page 2: in mathematical writing writing e.g. "298K" should be avoided in favour of "$2.98 \times 10^5$".
  - Page 2: the connection between number of parameters and dimensionality $D$ is not clear. Nothing stops an end user from applying a small model (with few parameters) to a high-dimensional dataset, after all.
- Page 2: Equation (2) is referenced alongside the fact that "dim(s) grows exponential wrt. dim(x)". However dim(s) refers to the dimensionality of the log-signature, whilst equation (2) is an expression for the signature. (I don't actually know the asymptotics for the log-signature -- in particular whether or not its exponential -- but it is certainly true that its gets quite large quite quickly.)
- Page 2: The notations dim(s) and dim(x) aren't defined.
- Table 1: "trianing" -> "training"
- Page 3: I think saying that a baseline is outperformed by "75%" isn't clear at all. 75% according to which metric, for one thing?
- Page 3: I believe it should be made clear that $\{x_i\}_{i=0}^N$ are a sample _along_ a time series, and not different samples down the batch dimension.
- Page 3: "e.g." (and similar abbreviations like "i.e.") are generally discouraged in academic writing (in favour of using the words "for example").
- Page 3: to be honest, I'd suggest avoiding comparison to the Fourier transform. At first glance these looks like they might be related -- they're both integrals of a path -- but that's about where the commonalities end. They're really very different things.
- Equation (2): $r_i$, $r_{i+1}$ aren't defined.
- Page 4: You have now introduced two notations for the same thing, namely $LogSig$ and $s_j$. I would suggest avoiding two different notations for the same object, for clarity's sake. If you absolutely must then make sure to link them with an explicit equality (and not merely the word "constitutes"), as again this maximises clarity.
- Page 4: I would suggest re-ordering the concepts that are introduced: NODEs, NCDEs, (Log-)signatures, NRDEs, LSTI. This helps to build up concepts in the order that will probably be most familiar to the reader.
- Page 4: NCDEs: I would note that NCDEs conversely have advantages that NRDEs do not. In particular NCDEs can use high-order numerical ODE solvers. (NRDEs are really just about choosing a particular differential equation solver -- namely a low-order one that happens to utilise higher terms in the log-signature.)
- Algorithm 1: I don't see that including this actually adds anything.
- Page 5: the constraint $dim(e(t)) = dim(logsig(...))$ doesn't actually seem to actually be necessary. Any low dimensionality would suffice.
- Equations (10)--(12): I note that a tanh is used before the final fully-connected layer. This was something of an oddity in Morrill et al. 2021; the NCDE formulation of Kidger et al. 2020 placed the tanh *after* the final fully-connected layer instead. As far as I know there's no reason to prefer one over the other, although the tanh-after formulation is more common. I would be curious if the authors have any observations on which is more effective?

**Summary Of The Paper:**

This paper suggests hybridising neural rough differential equations (NRDEs) with nonlinear dimensionality reduction techniques (autoencoders). This is an entirely reasonable approach, and by doing so the authors demonstrate impressive improvements across long time series benchmarks.

**Summary Of The Review:**

The main idea is sound and the empirical justification is impressively thorough. I have some concerns about clarity/novelty but these are not sufficient to detract from my overall positive impression of this paper.

---

### Official Review · Reviewer_RtnZ · 2021-10-30

**Correctness:** 3
**Technical Novelty And Significance:** 3
**Empirical Novelty And Significance:** 4
**Recommendation:** 6
**Confidence:** 4

**Main Review:**

__Strengths:__

- The idea of pretraining the learned features using an encoder/decoder architecture is interesting and a good contribution to the field even though the principle, relying on autoencoders for pretraining is not uncommon. That said, in this case the autoencoder is not balanced (i.e. it is not reconstructing the original input).
- Experiments are very thorough covering a wide range of applications and tasks. The empirical study is a useful contribution to this stream of research.
- The reported performance is good, validating the idea.

_Question:_

Have the authors considered end-to-end training? If no, why not and if yes, what were the reasons?

__Weaknesses:__

The writing is not very clear in some places. For example, the authors state that the “log-signature transform provides a basis itself” citing Kidger et al. 2019.
 - It is not mentioned what the log signature is a basis of.
 - I cannot find any such statement in the source. The only relevant statement I can find in Kidger et. al. relates to the signature (and its universal non-linearity) __not__ the log-signature.
Can the authors clarify what they mean?

_Minor:_

The first sentence is a bit awkward. It is just a long list of 15(!) citations (more than 50% of all sources), mostly from the last two years, to underscore the relevance of time series data.
First, time series analysis is a much older area, so the recency bias is odd. Second, a lot of the sources are unpublished. I feel like the references should be better motivated if they need to stay.

__Recommendation__:

In principle, the paper is a good contribution. However, the writing of the paper is sometimes unclear and I am reluctant to recommend acceptance so long as there are (mathematical) statements that I cannot verify. That said, I believe the authors can fix this.


**Summary Of The Paper:**

The authors propose to increase the efficiency of neural rough differential equations by encoding the log-signature into a lower dimensional space that is predictive of higher order signatures during a pre-training stage.
This enables the authors to work with lower dimensional log-signatures during the main training stages. The method performs competitively on a range of long time-series tasks.


**Summary Of The Review:**

The paper proposes an interesting extension to neural rough differential equations which shows good results in practice and is in principle a reasonable contribution to ICLR. However, the writing lacks detail in some instances (see comments) to an extent that the claims cannot be verified.
For this reason, I will stop short of recommending accepting at this point

---

### Official Review · Reviewer_GYq6 · 2021-10-31

**Correctness:** 4
**Technical Novelty And Significance:** 4
**Empirical Novelty And Significance:** 4
**Recommendation:** 8
**Confidence:** 4

**Main Review:**

I very much like the paper, particularly the main idea of using a pretrained signature-based autoencoder for time series. I believe this is novel and a natural improvement over standard techniques involving higher-depth (log-)signatures, which suffer from a "curse of dimensionality" with respect to depth. Whilst the proposed signature-based autoencoder does fit neatly with the NRDE model, I think that it is interesting for time series problems/models more generally.

The authors demonstrate that their methodology significantly (and consistently) outperforms pre-existing continuous-time models for long time series problems. These experiments are very thorough and compare against several baselines (ODE-RNN, NCDE, NRDE and the recently proposed Attentive NCDEs of [Jhin et al. 2021](https://arxiv.org/abs/2109.01876)) across 6 datasets.

Overall, I believe that this is a strong paper and the proposed methodology would be of great interest to the community. I could not find any substantial weaknesses, and thus only have the following minor points / typos to discuss:

### Minor points

* Autoencoders (and other dimensionality reduction techniques) are commonly used in machine learning. I believe this should be discussed in the introduction.

* I would say that the prevalent example of a control path X is a Wiener process coupled with time (i.e. dX_t = d(t, W_t)), which gives a stochastic differential equation. More generally, it should be clearer that time points $(t_i)$ can be included as an extra channel of the data $(x_i)$ when constructing the path $X$.

* In Section 2, I think NRDEs should be introduced after NCDEs and adjoint backpropagation should be mentioned (since one of the advantages of Neural ODE models is that they can be trained in a memory-efficient manner using an adjoint equation). Likewise, it is worth mentioning whether adjoint backpropagation is used in the "Training Method" and "Experimental Evaluation" sections.

* As I understand it, the autoencoder takes $\big(LogSig_{r_i, r_{i+1}}^{D_1}\big)$ as input and is trained using $\big(LogSig_{t_i, t_{i+1}}^{D_2}\big)$ where $D_1 < D_2$. So if the partition used to compute each $LogSig_{t_i, t_{i+1}}^{D_2}$ is finer than $(r_i)$, then some information in $\big(LogSig_{t_i, t_{i+1}}^{D_2}\big)$ may not be contained in $\big(LogSig_{r_i, r_{i+1}}^{D_1}\big)$. Thus, the claim that "the decoder should recover the higher-depth log-signature" on page 3 could be further discussed.

* I am not a fan of the notation $LogSig_{r_i, r_{i+1}}(X(t))$ on pages 5 and 6, since the log-signature transform takes a continuous path as input and $X(t)$ is a vector. Hence I prefer the notation $LogSig_{r_i, r_{i+1}}(X)$, as used in the original NRDE paper.

* $r_0, r_1$ is used instead of $r_i, r_{i+1}$ on pages 5 and 6. Moreover, I think $r_i, r_{i+1}$ should be detailed in Equations (8) and (16) in a similar manner to Equation (3) – that is, it should be clear that $t\in[r_i, r_{i+1})$.

* NRDE$_1$ is only theoretically identical to NCDE when piecewise linear interpolation is used to construct $X$ (page 7). For example, if the NCDE uses natural cubic splines, then the methods are slightly different.

* I like the authors use of tables on both pages 8 and 9. Perhaps Tables 2 and 3 could use bold text to highlight which models give the best improvements? Similarly, Table 4 could use bold for "best in column" and bold text + asterisk for "best over all models". (this comment is also relevant to the tables in the appendix)

### Typos

* I would say "continuous control path" instead of "control continuous path" (page 1)
* "calculated" instead of "calculate" (page 2)
* "Pre-training" instead of "Pre-trianing" (page 3)
* "create" instead of "crate" (page 7)
* The use of the word "Appendix" (pages 2, 6, 7, 8 and 10). For example, I believe "In Appendix" should be either "in the appendix" or followed by a letter – such as "In Appendix A"
* "sensitivity experiments" seems to have a lower case "s" (page 18 in appendix)
* "the encoder-decoder structure is more trained" (page 18 in appendix)

**Summary Of The Paper:**

The paper seeks to improve upon the recently introduced "Neural Rough Differential Equation" (NRDE) model by using an autoencoder to reduce the dimensionality of the input time series. The proposed autoencoder embeds "log-signature" information into a lower-dimensional space and can be trained prior to the main NRDE model. This approach is novel, easily understood, and intuitively should outperform previous log-signature based models. The authors demonstrate that it gives significant improvements on long time series problems through several experiments.

**Summary Of The Review:**

The LORD-NRDE model proposed by the paper is novel, intuitive and shows impressive performance across a wide varying of experiments. In particular, I find the use of a signature-based autoencoder for time series particularly interesting (in its own right). For these reasons, I would recommend the paper for acceptance into ICLR.

---

### Official Review · Reviewer_ncTs · 2021-11-02

**Correctness:** 4
**Technical Novelty And Significance:** 3
**Empirical Novelty And Significance:** 2
**Recommendation:** 8
**Confidence:** 4

**Main Review:**

The manuscript is well presented and clear, and the details on experiments are thorough in the main text and would be easy to reproduce, the appendix is also clear and to the point.

The problem of reducing spatial overhead in signature methods is an important one and is in need of more study, and I believe the manuscript is a fine addition the the literature. I would also like to point the authors to the following paper:
https://openreview.net/forum?id=dx4b7lm8jMM
which tackles the same problem by a different method.

In the appendix it looks like the autoencoder is trained for as many or many more iterations than the NRDE for a large share of the tasks,  but I can't find any mention on how it affects the training time.

The proposed model is specifically made for long time-series, but not all the baseline methods are. In the interest of fairness it would be nice if there was some mention (in the appendix) of how well the model compares to the baselines on some shorter time-series too.

Signature and neural CDE methods in ML are up-and-coming fields and the method proposed here is an interesting take on dimensionality reduction for these methods which performs very well compared to the baselines chosen. I think the manuscript is of high quality and believe it should be accepted.


**Summary Of The Paper:**

The manuscript proposes a new method of training neural RDEs with a pretrained autoencoder to reduce overhead of the log-signature transform when dealing with long time-series. The neural RDE is trained on the image of a pretrained autoencoder as a way to reduce complexity and memory usage.

The methods is compared to several different baselines, including neural CDE and ODE methods. These are tested on 6 real world datasets for both classification and forecasting.

**Summary Of The Review:**

The ideas presented in the manuscript are novel and well presented, and the method seems to drastically cut down on overhead, and will be useful for both classification and forecasting.

---

### Decision · Program_Chairs · 2022-01-20

**Decision:**

Accept (Poster)

**Comment:**

This paper proposes a novel method for training neural rough differential equations, a recent model for processing very long time-series data. The method involves a lower-dimensional embedding of the log-signature, which is obtained via pretrained autoencoder to reduce overhead. The results show significant and consistent improvements over previous methods on long time-series data.

Overall, the reviewers and I all agree that this paper offers a novel and impactful contribution leading to significant improvements over previous state-of-the-art methods for training neural rough differential equations. I recommend acceptance.